# Integrated proteogenomic and metabolomic characterization of papillary thyroid cancer with different recurrence risks

Ning Qu[1,2,10], Di Chen[3,10], Ben Ma[1,2,10], Lijun Zhang[4,5], Qiuping Wang[3], Yuting Wang[1,2], Hongping Wang[6], Zhaoxian Ni[1,2], Wen Wang [3], Tian Liao[1,2], Jun Xiang[1,2], Yulong Wang [1,2], Shi Jin[7], Dixin Xue[8], Weili Wu[8], Yu Wang[1,2] ✉, Qinghai Ji[1,2] ✉, Hui He[1,2,7] ✉, Hai-long Piao [3,9] ✉ & Rongliang Shi[1,2] ✉

Although papillary thyroid cancer (PTC) has a good prognosis, its recurrence rate is high and remains a core concern in the clinic. Molecular factors contributing to different recurrence risks (RRs) remain poorly defined. Here, we perform an integrative proteogenomic and metabolomic characterization of 102 Chinese PTC patients with different RRs. Genomic profiling reveals that mutations in *MUC16* and *TERT* promoter as well as multiple gene fusions like *NCOA4-RET* are enriched by the high RR. Integrative multi-omics analyses further describe the multi-dimensional characteristics of PTC, especially in metabolism pathways, and delineate dominated molecular patterns of different RRs. Moreover, the PTC patients are clustered into four subtypes (CS1: low RR and BRAF-like; CS2: high RR and metabolism type, worst prognosis; CS3: high RR and immune type, better prognosis; CS4: high RR and BRAF-like) based on the omics data. Notably, the subtypes display significant differences considering BRAF and TERT promoter mutations, metabolism and immune pathway profiles, epithelial cell compositions, and various clinical factors (especially RRs and prognosis) as well as druggable targets. This study can provide insights into the complex molecular characteristics of PTC recurrences and help promote early diagnosis and precision treatment of recurrent PTC.

Thyroid cancer (TC) is the most common malignant tumor of the endocrine system, and papillary thyroid cancer (PTC) is the most common type of thyroid malignancy. Although PTC is in general with low-grade malignancy and favorable long-term prognosis, the recurrence rate is relatively high with up to 20% of PTC patients having recurrences[1,2]. The American Thyroid Association (ATA) risk stratification system categorizes the recurrence risks (RRs) into low, intermediate, and high levels based on several recurrence-relevant clinical

[1]Department of Head and Neck Surgery, Fudan University Shanghai Cancer Center, Shanghai, China. [2]Department of Oncology, Shanghai Medical College, Fudan University, Shanghai, China. [3]Dalian Institute of Chemical Physics, Chinese Academy of Sciences, Dalian, China. [4]Department of General Surgery, Ganmei Affiliated Hospital of Kunming Medical University (The First People's Hospital of Kunming), Kunming, Yunnan, China. [5]Department of Surgery, Kunming Medical University, Kunming, Yunnan, China. [6]Department of Endocrinology, Putuo Hospital, Shanghai University of Traditional Chinese Medicine, Shanghai, China. [7]Department of Laparoscopic Surgery, The First Affiliated Hospital of Dalian Medical University, Dalian, China. [8]Department of Thyroid and Breast Surgery, The Third Affiliated Hospital of Wenzhou Medical University, Wenzhou, China. [9]Department of Biochemistry & Molecular Biology, School of Life Sciences, China Medical University, Shenyang, China. [10]These authors contributed equally: Ning Qu, Di Chen, Ben Ma. ✉e-mail: neck130@sina.com; jq_hai@126.com; hehui@dmu.edu.cn; hpiao@dicp.ac.cn; shirongliang@126.com

factors[1]. Uncovering molecular factors associated with PTC RR may promote early detection of PTC recurrences and better treatment. The elevated serum levels of thyroglobulin (Tg) were found to be associated with PTC recurrence and have been applied for recurrence surveillance in clinical use[3]. Some recurrence-relevant genes and microRNAs were identified based on the transcriptomics data[4,5]. However, the molecular basis underlying different RRs are still not fully revealed.

High-throughput omics methods have been applied to explore the molecular atlas of PTC[4,6–10]. Accordingly, the molecular landscape of PTC has been described. The high mutation frequencies in *BRAF*, *RAS*, *TERT* promoter, and gene fusions involving *RET* have been widely observed[10,11]. Proteomics and metabolomics studies described the remarkably altered protein and metabolite profiles in PTC[7,9]. Transcriptomics-based analysis also identified various metabolic enzymes that play key roles in PTC[12–15]. Existing omics-based studies manifest that PTC is molecularly complex, and the molecular characteristics underlying PTC recurrence require further, more in-depth integrative investigations.

Here, we aim to obtain a more comprehensive perspective on the molecular landscape of PTC with different RRs. To do this, we perform an integrated proteogenomic and metabolomic investigation of PTC of 102 Chinese PTC patients. Our integrated analysis describe the complicated and distinctive molecular features of the PTC patients and identify the RR-relevant molecular landscape from the genomic, transcriptional, proteomic and metabolism perspectives. We also redefine four molecular subtypes of PTC which not only possess distinctive molecular characteristics but also show significant differences in clinical and pathological scales, especially for RR patterns and recurrence-free prognosis. This multi-omics study holds immense potential in offering valuable data resources for unraveling the intricate molecular mechanisms of PTC recurrences, and the redefined molecular subtypes can significantly contribute to enhancing precision diagnosis and treatment of recurrent PTCs, thus leading to improved long-term survival rates.

## Results

### Overview of the multi-omics study of PTC

A total of 102 PTC patients were collected (Supplementary Data 1). The average age at diagnosis was 42 years (range, 15–77), with 63.73% females ($n = 65$). There were respectively 47.06% high RR ($n = 48$), 27.45% intermediate RR ($n = 28$) and 25.49% low RR ($n = 26$) patients. The clinicopathological characteristics were summarized in Table 1 (see also Supplementary Data 1).

To describe the molecular landscape of PTC, multi-omics data, including genomics, transcriptomics, metabolomics, proteomics, and phosphorylated (phospho)-proteomics, were performed (Supplementary Fig. S1a). Whole exome sequencing (WES)-based genomics data were from 97 tumor tissue samples and 33 paired normal tissues, the RNA-sequencing (RNA-seq)-based transcriptomics data (16,925 genes) were from 92 tumor tissue samples and 34 paired normal tissue samples, metabolomics profiling (503 metabolites) were conducted on 102 tumor tissue samples and 37 paired normal tissue samples, and proteomics (3147 proteins) and phospho-proteomics (652 phospho-proteins) profiling were performed on 37 paired tumor-normal tissues (Supplementary Fig. S1a and Supplementary Data 1).

### Genomic profiling of the PTC patients

An average of 74 nonsynonymous somatic point mutations and 2 indels were identified in the 97 Chinese PTC patients. Consistent with most genome studies about PTC[10,16], the most frequent somatic mutation gene was *BRAF* (47%, all belong to V600E mutation, Fig. 1a). In addition, frequently mutated cancer-associated genes also included *MUC16* (36%), *RNF213* (8%), and *MSH6* (7%) (Fig. 1a), showing higher mutation frequencies than the cancer genome atlas (TCGA) PTC

**Table 1 | Clinicopathological characteristics of the collected samples ($n = 102$)**

| Feature | Statistics | |
|---|---|---|
| Age (year) | | 41.48 (±15.11) |
| Gender | Female | 65 (63.73%) |
| | Male | 37 (36.27%) |
| RR | High | 48 (47.06%) |
| | Intermediate | 28 (27.45%) |
| | Low | 26 (25.49%) |
| Tumor size (cm) | | 3.254 (±1.872) |
| ETE | TRUE | 32 (31.37%) |
| | FALSE | 70 (68.63%) |
| ENE | TRUE | 32 (31.37%) |
| | FALSE | 70 (68.63%) |
| LNM | TRUE | 88 (86.27%) |
| | FALSE | 14 (13.73%) |
| LNM.No | | 8.931 (±7.669) |
| LNM.3cm | TRUE | 85 (83.33%) |
| | FALSE | 17 (16.67%) |
| T stage | T1 | 28 (27.45%) |
| | T2 | 33 (32.35%) |
| | T3 | 29 (28.43%) |
| | T4 | 12 (11.76%) |
| TNM stage | I | 74 (72.55%) |
| | II | 19 (18.63%) |
| | III | 1 (0.9804%) |
| | IV | 8 (7.843%) |
| N stage | N0 | 14 (13.73%) |
| | N1a | 19 (18.63%) |
| | N1b | 69 (67.65%) |
| M stage | M0 | 87 (85.29%) |
| | M1 | 15 (14.71%) |

*ETE* extrathyroidal extension, *ENE* extranodal extension, *LNM* lymph node metastasis, *LNM.No* number of metastatic lymph nodes, *LNM.3cm* metastatic lymph node size larger than 3 cm.

dataset[10] (Supplementary Fig. S1b). Here, the *MUC16* mutations were specifically enriched in the PTC patients with high RR (Fig. 1b), and also associated with multiple pathological factors, including high RR ($P = 0.027$), recurrence ($P = 0.010$), metastatic lymph node size larger than 3 cm (LNM.3cm) ($P = 0.018$), T3 stage ($P = 0.032$), N1b stage ($P = 0.0061$) and M1 stage ($P = 0.021$) (Fig. 2c, examined by hypergeometric distribution). Meanwhile, this Chinese PTC cohort did not contain mutations in RAS which was mutated in 13% of samples for the TCGA-PTC cohort[10], but the mutation frequency was about 4.1–6.0% in the other Chinese cohorts[16,17].

There were also frequent TERT promoter mutations (C228T, 14%) in the PTC patients. The mutations were also significantly enriched in the high RR patients (Fig. 1b, d), and frequently overlap with certain pathological or clinical factors including high RR ($P = 0.0026$), recurrence ($P = 0.0030$), LNM.3cm ($P = 0.011$), extrathyroidal extension (ETE) ($P = 0.0013$), lymph node metastasis (LNM) ($P = 0$), or extranodal extension (ENE) ($P = 0.0077$) (Fig. 1d, examined by hypergeometric distribution).

Gene rearrangements in *RET*, *NTRK* and *BRAF* have been frequently identified in PTC[18]. Here, *RET* fusions (*CCDC6-RET* 8%, *NCOA4-RET* 5%) were the most frequent fusions, and multiple *NTRK* fusions (*NTRK3-ETV6*, *TPR-NTRK1*, *ETV6-NTRK3*) were also identified (Fig. 1a and Supplementary Fig. S1c). In addition, several other gene fusions (*FBXO25-SEPTIN14*, *TLK2-FAM157A*, *ZNF33B-NCOA4*) showing rare

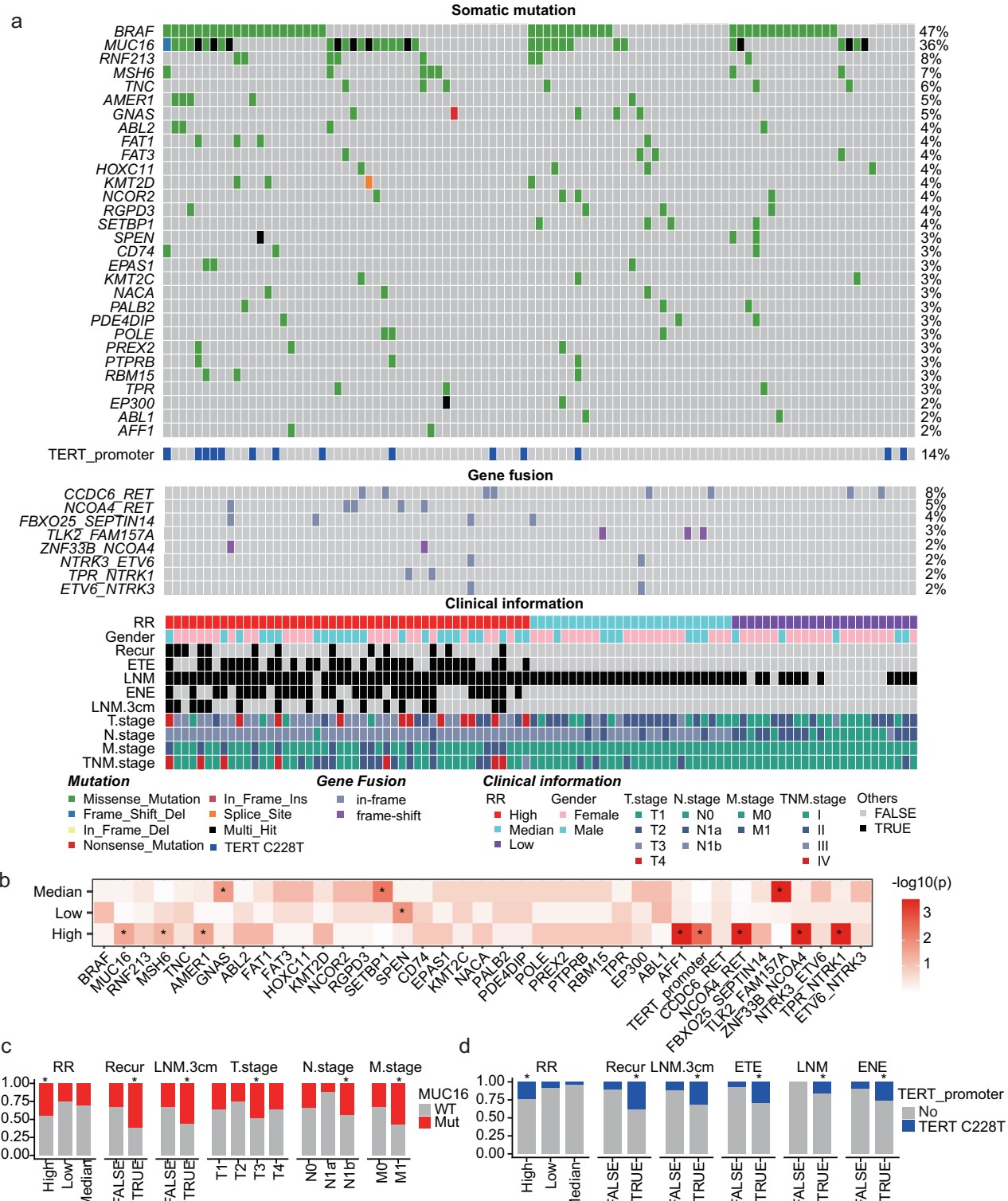

**Fig. 1 | Genetic profile of the PTC patients with different RRs. a** Genetic profile and associated clinical information of 97 PTC patients. **b** Mutations with significant enrichment in one type of RR. *$P < 0.05$, examined by hypergeometric distribution, one-sided. **c, d** Clinical features showed significant enrichment in the samples with mutations in *MUC16* (**c**) or *TERT* promoter (**d**). The bar length represents the percentage of samples with mutated (**c**: red bar, **d**: blue bar) or wild type (WT) (gray bar) genes. *$P < 0.05$, examined by hypergeometric distribution, one-sided. RR recur risk, LNM.3cm metastatic lymph node size larger than 3 cm, ETE extra-thyroidal extension, LNM lymph node metastasis, ENE extranodal extension. Source data are provided as a Source Data file.

frequencies in previous PTC studies were also identified (Fig. 1a). Interestingly, several gene fusions (*NCOA4-RET*, *TLK2-FAM157A*, *ZNF33B-NCOA4*, *TPR-NTRK1*) also showed specific enrichment in the high RR (Fig. 1b).

## Multi-omics-based comparison of tumor and normal tissues of PTC patients

In addition to the genomic alterations, differentially expressed molecules (DEMs) were recognized by comparing between tumor and

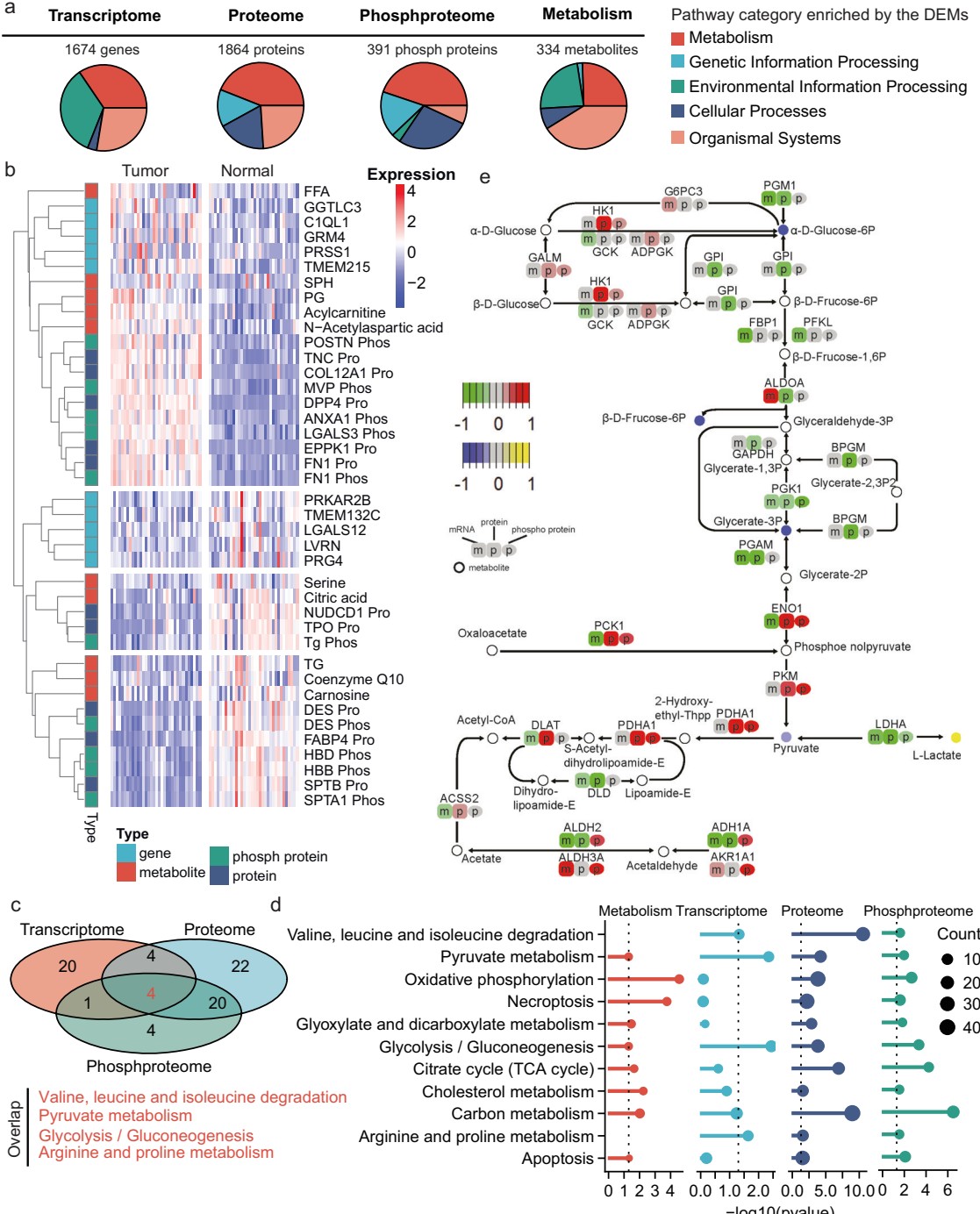

**Fig. 2 | Multi-omics-based profiling of the tumor and normal thyroid samples.**
**a** Pathway categories enriched by the DEMs. **b** Heatmap showing the top-rank DEMs. Only molecules with the top-10 significant *P* values for each omics type are listed. The metabolite levels for TG, FFA, PG, and SPH were the summarized abundances of different kinds of TG, FFA, PG, and SPH. The suffix Pro and Phos, respectively, represent proteins and phosphoproteins. DESeq2 (two-sided) was applied to find the differentially expressed genes, and the Wilcox test (paired, two-sided) was applied to find the differentially expressed metabolites, proteins and phosphoproteins. *P* values were adjusted by Benjamini−Hochberg method. **c** Venn diagram of the pathways respectively enriched by the DEMs determined by transcriptomics, proteomics and phospho-proteomics. **d** Pathway enrichment results. Only pathways showing significance in at least three types of omics are shown. Enrichment was examined by hypergeometric distribution, one-sided, and *P* values were adjusted by the Benjamini− Hochberg method. **e** The multi-omics differences between tumor and normal samples considering the glycolysis pathway. Source data are provided as a Source Data file.

matched normal samples based on the multi-omics profiling data (Supplementary Fig. S2). As a result, four types of DEMs, including 1674 genes ($P < 0.01$, DeSeq2[19]), 1864 proteins ($P < 0.01$, Wilcox test, paired), 391 phosphoproteins ($P < 0.01$, Wilcox test, paired) and 334 metabolites ($P < 0.01$, Wilcox test, paired) were recognized (Fig. 2a). Increased

FFAs in PTC tumors have also been identified by other metabolomics studies[20]. Proteins like tenascin (TNC), fibronectin 1 (FN1), dipeptidyl peptidase 4 (DPP4), and phosphoproteins of major vault protein (MVP) and FN1 showed remarkable upregulation in the PTC tumor tissues, while proteins thyroid peroxidase (TPO), desmin (DES) and fatty acid

binding protein 4 (FABP4), and phosphoproteins of thyroglobulin (Tg), DES, hemoglobin subunit delta (HBD) and hemoglobin subunit beta (HBB) were downregulated in the PTC tumors (Fig. 2b). TNC was reported to show remarkably high expressions in medullary TC[21], here, we found it was also upregulated in the PTC tumors. The upregulation of FN1 was observed in all types of TC[22]. TPO, an essential enzyme for the production of thyroid hormones, is expressed mainly in normal thyroid cells[23]. The expression levels of TPO were decreased in the PTC tumors when compared to the normal ones.

Pathways enriched by the four types of DEMs were identified respectively ($P < 0.05$, Hypergeometric Distribution). A large fraction of the enriched pathways were metabolism pathways even for the DEMs in terms of genes, proteins and phosphoproteins (Fig. 2a), and the pathways enriched by genes, proteins and phosphoproteins simultaneously all fell in metabolism pathways including valine, leucine, and isoleucine degradation, pyruvate metabolism, glycolysis/gluconeogenesis, as well as arginine and proline metabolism (Fig. 2c), where glycolysis and pyruvate metabolism were also enriched by the differentially expressed metabolites (Fig. 2d). Meanwhile, multiple metabolism pathways like oxidative phosphorylation and citrate cycle were enriched by at least three types of DEMs (Fig. 2d). These together suggest the remarkable metabolic alterations in PTC tumor tissues.

The multi-omics-based pathway analysis enable a comprehensive description of the pathway alteration. Taken glycolysis as one example (Fig. 2e), we observed that although most enzymes were downregulated considering the mRNA expressions (e.g., *HK1*, *PGM1*), some of them were upregulated in the protein or phospho-protein levels (e.g., ENO1, PCK1, PDHA1). Meanwhile, the metabolite changes were mainly reflected in the reduced levels of glucose, fructose, glycerate-3P and pyruvate and increased levels of lactate (i.e., L-Lactate) (Fig. 2e). Increased levels of lactate in TC and many other cancer types have been widely reported[8]. The multi-omics-based pathway alteration patterns help further explain potential mechanisms, including alteration of the direct enzyme LDHA (in both mRNA and protein levels) and associated up-/downstream changes (e.g., ENO1, PKM, and PDHA1).

In addition to the metabolism pathways, two cell death-relevant pathways, necroptosis and apoptosis, were also significantly enriched by the DEMs in terms of metabolites, proteins and phosphoproteins (Fig. 2d). Most of the DEMs in the necroptosis and apoptosis pathways were upregulated in the PTC tumor tissues than the normal tissues (Supplementary Fig. S3a, b).

Based on these multi-omics data, we also observed the potential impacts of the three most frequent mutations *BRAF*, *MUC16*, and *TERT* promoter on the multi-omics profiles (Supplementary Fig. S4a, b). Enrichment analyses showed these mutations were also related with alterations in multiple metabolism pathways, especially for BRAF (Supplementary Fig. S4c). For example, the differentially expressed mRNAs between PTC samples with and without BRAF mutations were significantly enriched by glycerolipid metabolism and biosynthesis of amino acid pathways (Supplementary Fig. S4c).

**Multi-omics-based molecular features of PTC with different RRs**
The molecular expression features underlying different RRs of PTC were also characterized (Fig. 3a). The high RR PTC patients showed higher expression levels in multiple lipids like TGs, FFAs and the other metabolites like histamine and kynurenine. The high RR also displayed higher expressions in genes like *MMP13, CST1, COL11A1*, proteins like Tg, PTRRG, VWA1 and phosphoproteins like EPPK1, ALDH1A1, and LAMC1. The intermediate RR PTC patients showed higher expressions in metabolites like several FFAs and kynurenine, genes like *IGFN1, LOC391322,* and *ZNRD1*, proteins like FTL, FABP5, and APOB, and phosphoproteins like C1QB, HBB, and HBD. The low RR patients showed higher expressions in metabolites like PG (18:2_18:2) and OAHFA (18:2_18:1), genes like *JSRP1, TCAP,* and *TNNI2*, proteins like ACADL, ABHD11 and FN1 and phosphoproteins like TNC,

FN1, and POSTN. Compared to the alterations between tumor and normal samples, the high RR PTC tumor samples showed reversed alterations compared to intermediate or low RR ones considering various molecules (Fig. 3a and Supplementary Fig. S5a–d). For instance, although the PTC tumor samples showed significantly reduced protein levels in Tg compared to the normal samples (Supplementary Fig. S5c), the high RR PTC samples were with higher protein expressions of Tg compared to other tumor samples (Fig. 3a).

The expression profiles of the RR-relevant molecules, especially for the FFAs (FFA 26:2, FFA 24:2, FFA 26:4) and several proteins or phosphoproteins were highly associated (Fig. 3b, spearman correlation >0.65 or Spearman correlation <−0.65). The FFA 26:2, Tg, FN1, and 5-Lipoxygenase (ALOX5), phospho-FN1, and phospho-TNC harbored a relative hub position in the correlation network, suggesting their crucial roles in interactive regulations or signaling communications. ALOX5, as a non-heme iron-containing enzyme, can catalyze the peroxidation of polyunsaturated fatty acids[24]. Aberrant expression of ALOX5 has been observed in various types of cancers, including PTC[25]. Here, we also found ALOX5 showed specific low expressions in high RR PTC patients, and its alterations were associated with changes in many FFAs (Fig. 3b–d).

The different RRs also displayed remarkable differences in the pathway profiles. For high RR, the metabolites in various metabolism pathways, e.g., biosynthesis of amino acids and glycolysis, were upregulated, while the protein levels of metabolic enzymes were mainly downregulated (Fig. 3c). Except of the direct metabolism enzymes, there were other proteins showing remarkable associations with metabolite changes in PTC (Fig. 3f), e.g., protein Tg, phosphoprotein MSN (Fig. 3g–h). For the other pathways, the high RR showed upregulations in PI3K-AKT and TGF-beta signaling pathways (based on the mRNA expressions) and thyroid hormone synthesis (based on the protein expressions) (Supplementary Fig. S5e).

**Integrative correlation analysis of the multi-omics data**
The correlations between different types of omics data were evaluated based on a supervised multi-omics integrative analysis method called DIABLO (Data Integration Analysis for Biomarker discovery using Latent cOmponents)[26] which can simultaneously maximize the correlations among different types of omics and identify key molecules which can discriminate different sample groups (i.e., high, intermediate and low RR groups and the normal sample group). As a result, the general correlations between metabolism, proteomics, and phospho-proteomics were high (no less than 0.88), suggesting common information among metabolism, proteomics and phosphoproteomics. The proteomics are expected to be highly correlated with the transcriptomics, according to the central dogma that the information pasts from DNA to RNA to protein. However, relatively low correlations were observed between transcriptomics and proteomics or phospho-proteomics (Fig. 4a). From the basis, the mRNA−protein correlations were low, with the sample-wise and gene-wise median mRNA−protein Spearman correlation coefficients, respectively, 0.29 and 0.086 (Supplementary Fig. S6a, b). From the perspective of pathways, the genes/proteins of metabolism pathways showed a relatively higher correlation (but still around 0.25–0.5) than other pathways, while the genes/proteins involved in pathways with large protein complexes like ribosome, spliceosome, mRNA surveillance, and autophagy, displayed low or even opposite correlations (around −0.25 to 0) (Supplementary Fig. S6c, Supplementary Data 2, Kolmogorov−Smirnov test, one-sided). Similar results were also reported by previous studies[21,27,28]. Besides, dysregulation in post-transcriptional modifications, like ubiquitination enzymes (HUWE1, CUL4A, TRIM25, etc.) and deubiquitination enzymes (USP7, USP10, USP24, etc.) in these PTC tumor samples (Supplementary Fig. S6d) may also lead to the low consistency.

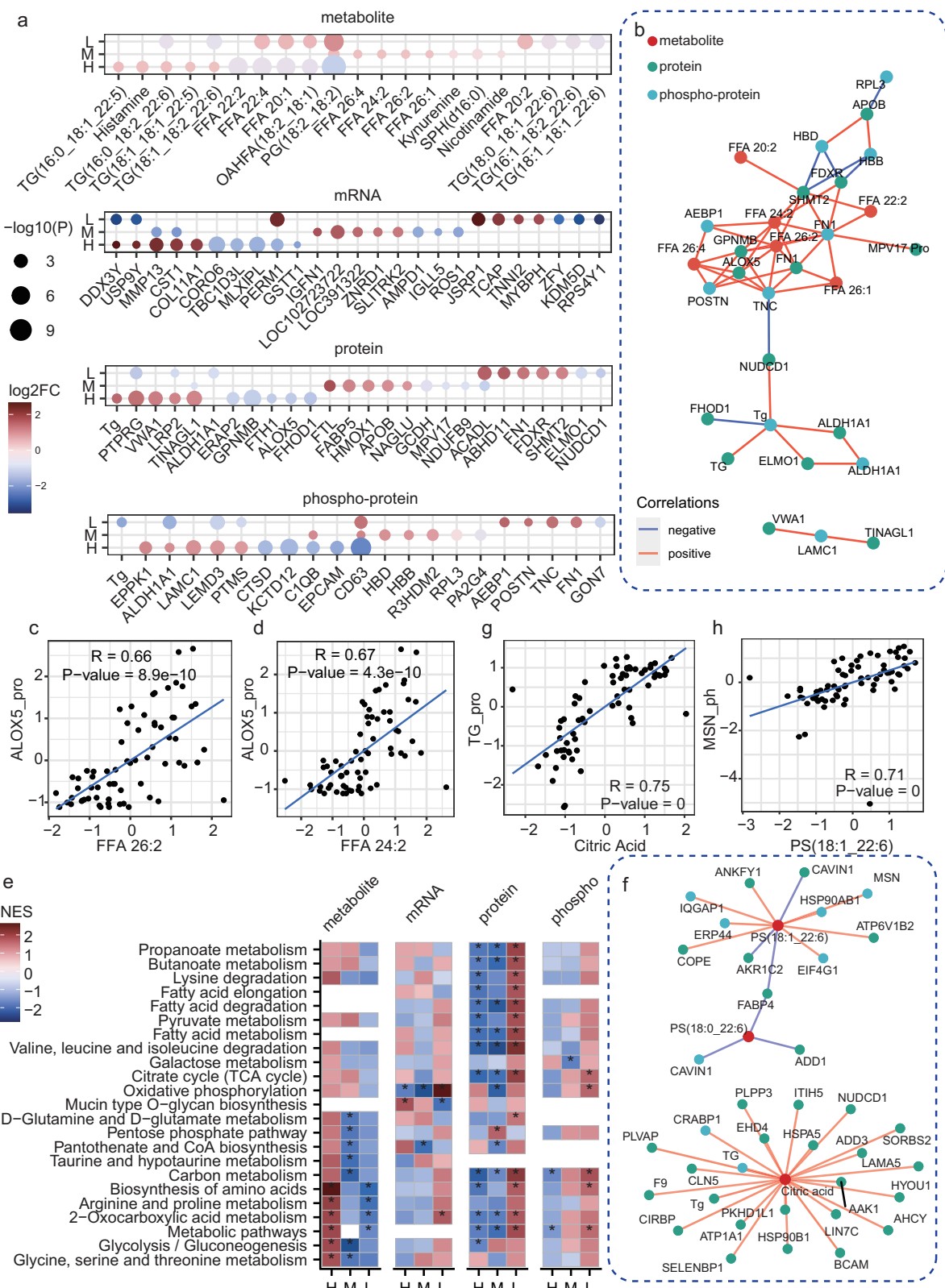

The key molecules were further clustered into four network modules based on their expression profiles and inter-correlations, and different modules showed distinctive expression profiles (Fig. 4b) and interaction patterns (Fig. 4c). The molecules in the first module (M1) were mainly composed of extracellular matrix (ECM) relevant proteins including FBLN5, NID1, NID2, COL4A2, TINAGL1, VWA1, and different chains of the laminin proteins (LAMA4, LAMB1,

LAMC1)[29], they showed high inter-correlations and possessed higher expression levels in the high RR groups than the other tumor samples (Fig. 4b, c), highlighting the key role of ECM interactions in PTC recurrence. The second module (M2) was composed of multiple metabolism-relevant phosphoproteins like PGK1, PSMF1, PRDX1, and TOM1, they showed lower expressions in the tumor tissues, especially the high RR tumor tissues (Fig. 4b). Their expressions were also

**Fig. 3 | Multi-omics landscape of different recurrent risk. a** Molecules with significant differential expressions in patients with different RR. H high RR, M intermediate RR, L low RR. The difference was examined by the Wilcox test, two-sided. *P* values were adjusted by the Benjamini–Hochberg method. **b** Correlation network of the molecules showed significant differential expressions among different RRs. **c**, **d** Scatter plot showing the correlations (Spearman correlation, two-sided) between protein ALOX5 and metabolite FFA 26:2 (**c**) and FFA 24:2 (**d**). **e** Metabolism pathway enrichment results of the RR-relevant molecules. NES normalized enrichment scores. *\*P* < 0.05, by gene set enrichment analysis (GSEA, Kolmogorov–Smirnov test, one-sided). **f** Correlation network of the molecules. Only metabolites in the downregulated metabolism pathways for the high-risk samples concerning proteomics were shown. The displayed edges are with correlation coefficients larger than 0.5. **g**, **h** Scatter plot showing the correlations (Spearman correlation, two-sided) between metabolite citric acid and protein TG (**g**), metabolite PS (18:1_22:6), and phospho-protein MSN (**h**). Source data are provided as a Source Data file.

associated with the proteins IGKV3-15, RPS27A, genes *MYH11*, and *HTR2A* (Fig. 4c). The third module (M3) was the largest module, and molecules in M3 mainly showed higher expressions in the tumor tissues (regardless of the RRs) than the matched normal tissues (Fig. 4b). There were three sub-modules in M3 which were aggregated by metabolites, genes and proteins/phosphoproteins (Fig. 4c, M3). The inter-correlated metabolites in M3 were mainly lipids, including phosphatidylcholines (PCs), phosphatidylethanolamines (PEs) and sphingomyelins (SMs). A large fraction of the proteins/phosph-proteins were involved in autophagy (STAT3/ATP6V1B2/ATP6V1C1/HGS/LAMTOR2/VPS13C/HSP90AA1)[30]. The genes were involved in glutathione metabolism (*GGTLC3/GGT2*)[30], immune response (*IFNE/PRSS2*)[30] and exocytosis secretion of thyroid-stimulating hormone (*TRHR*). Meanwhile, *TRHR, C1QL4* and *AMY1B* possessed inter-connection positions in the network of M3. Molecules in the fourth module (M4) mainly showed higher expressions in the intermediate and low RR groups (Fig. 4b). Metabolites including FFAs, Diacylglycerols (DGs) and Phosphatidylglycerols (PGs) and the fatty acid binding protein FABP5 formed an intermediated layer linking the genes and proteins or phosphoproteins in the network of M4, and multiple proteins and phosphoproteins (ERAP2/CYBB/CD74/DYNC1H1/RAB7A) in M4 were involved in antigen processing and presenting[30] (Fig. 4c), indicating the potential interactions between fatty acid metabolism and adaptive immune functions in PTC.

## Integrative stratification of PTC patients into four subtypes based on transcriptomics and metabolomics

Notably, although most high RR samples showed low expressions of molecules in M4, part of them also show similar expression profiles with the intermediate and low RR samples (Fig. 4b), implying alternative molecule subtypes different from the ATA risk classification may exist. We re-stratified the PTC patients into four subtypes based on a consensus integrative clustering analysis (see "Methods", Supplementary Fig. S7a) of the transcriptomics and metabolomics data (proteomics and phospho-proteomics were not considered here since the two types of omics were highly associated with metabolomics).

The four redefined subtypes showed significant differences in the transcriptional and metabolism profiles (Fig. 5a) as well as multiple clinical and mutation features, including RR ($P = 1.61 \times 10^{-5}$, Chi-square test), T stage ($P = 4.17 \times 10^{-2}$, Chi-square test), N stage ($P = 2.45 \times 10^{-3}$, Chi-square test), BRAF mutation ($P = 5.67 \times 10^{-3}$, Chi-square test), TERT promoter mutation ($P = 3.49 \times 10^{-3}$ for overlap between CS4 and TERT promoter mutation, examined by hypergeometric distribution) (Fig. 5a, b) and ENE ($P = 6.95 \times 10^{-3}$, Chi-square test, Supplementary Fig. S7b). The subtype CS1 contained more low RR patients, while the other three subtypes CS2 to CS4 had more high RR patients (Fig. 5b). The subtype CS2 and CS3 had more T3 stage, N1b stage patients but had less BRAF and TERT promoter mutations (Fig. 5b). The subtype CS4 was significantly enriched by BRAF and TERT promoter mutations ($P = 0.007227$ for BRAF, $P = 0.02937$ for TERT, Chi-squared test, Fig. 5b). Moreover, the four subtypes possessed different prognosis outcomes in terms of recurrence-free survival (Fig. 5c), where the subtype CS2 and CS3 respectively showed the worst and best prognosis among the three

high RR-enriched subtypes (Fig. 5d, e). By contrast, the prognosis differences based only on the RRs were not significant (Supplementary Fig. S7c). Taken together, the transcriptomics and metabolomics data help redefine four meaningful PTC subtypes.

## Multi-dimensional characterization of the four PTC subtypes

The four subtypes were with remarkably distinctive molecular profiles, and each subtype possessed various specifically up- or downregulated genes (Fig. 6a) and metabolites (Fig. 6b). Plasminogen (PLG) was reported to show significantly lower expressions in serum samples of PTC patients than the nodular goiter patients[31]. Here, the mRNA expression levels of *PLG* were higher in the low RR-dominated subtype CS1 than the other subtypes (Fig. 6a). *HSP6A* was found to be a potential biomarker to predict the prognosis of TC[32]. *ECM1* is associated with tumor invasiveness and poor prognosis in various cancer types[33]. Both *HSP6A* and *ECM1* showed CS2-specific higher expressions (Fig. 6a). Considering metabolites, the subtype CS1 had higher levels in FFAs, PGs, Fructose 1,6-diphosphate, etc.; the subtype CS2 had higher levels in stachydrine; the subtype CS3 had higher levels in TGs, citric acid, etc.; while the subtype CS4 showed higher levels in acyl-carnitines, adenosine, histamine, etc. (Fig. 6b).

In addition to the molecular features, the four subtypes also showed differences in the other key aspects, including tumor sizes, number of metastatic lymph nodes, tumor differentiation scores (TDSs)[34], BRAF-scores and RAS-scores. The subtype CS2 and CS3 showed larger tumor sizes than the subtype CS1 (Fig. 6c). The subtype CS1 had less number of metastatic lymph nodes than the other subtypes (Fig. 6d). The subtype CS2 showed higher TDSs than the subtype CS1 and CS4, and the subtype CS3 had higher TDSs than the subtype CS1 (Fig. 6e). Moreover, both subtype CS2 and CS3 showed higher RAS scores and lower BRAF scores comparing to the subtype CS1 and CS4 (Fig. 6f, g). Correspondingly, the subtypes CS2 and CS3 had lower BRAF mutation frequencies than CS1 and CS4 (Fig. 5b).

Furthermore, the pathway profiles for the four subtypes were also identified. Although no significant differences in terms of the tumor sizes, number of metastatic lymph nodes, TDSs, BRAF and RAS scores were observed between the subtype CS2 and CS3 (Fig. 6c–g and Supplementary Fig. S8a), they displayed noteworthy opposite trends in the pathway profiles (Fig. 6h, i). For the metabolism pathways, the mRNA expressions of enzymes in various metabolism pathways were upregulated for CS2 and downregulated for CS3 (Fig. 6h). Reversely, most immune-relevant pathways were upregulated for CS3 but downregulated for CS2 (Fig. 6i). The upregulation in various metabolism enzymes in CS2 imply a high metabolite consumption and can partly explain why few metabolites show higher levels in the CS2 subtype (Fig. 6b).

Single-cell RNA-seq (scRNA-seq) analysis has been performed to illustrate the tumor microenvironment heterogeneity of the PTC samples[35]. We used a deconvolution method to predict the tumor microenvironment compositions of the PTC tumor samples based on the bulk sample RNA-seq data in our study and a previously reported scRNA-seq dataset of PTC[35] (see also "Methods"). As a result, the four subtypes also displayed distinctive cell compositions, especially for the epithelium sub-populations (Fig. 6j and Supplementary Fig. S8b),

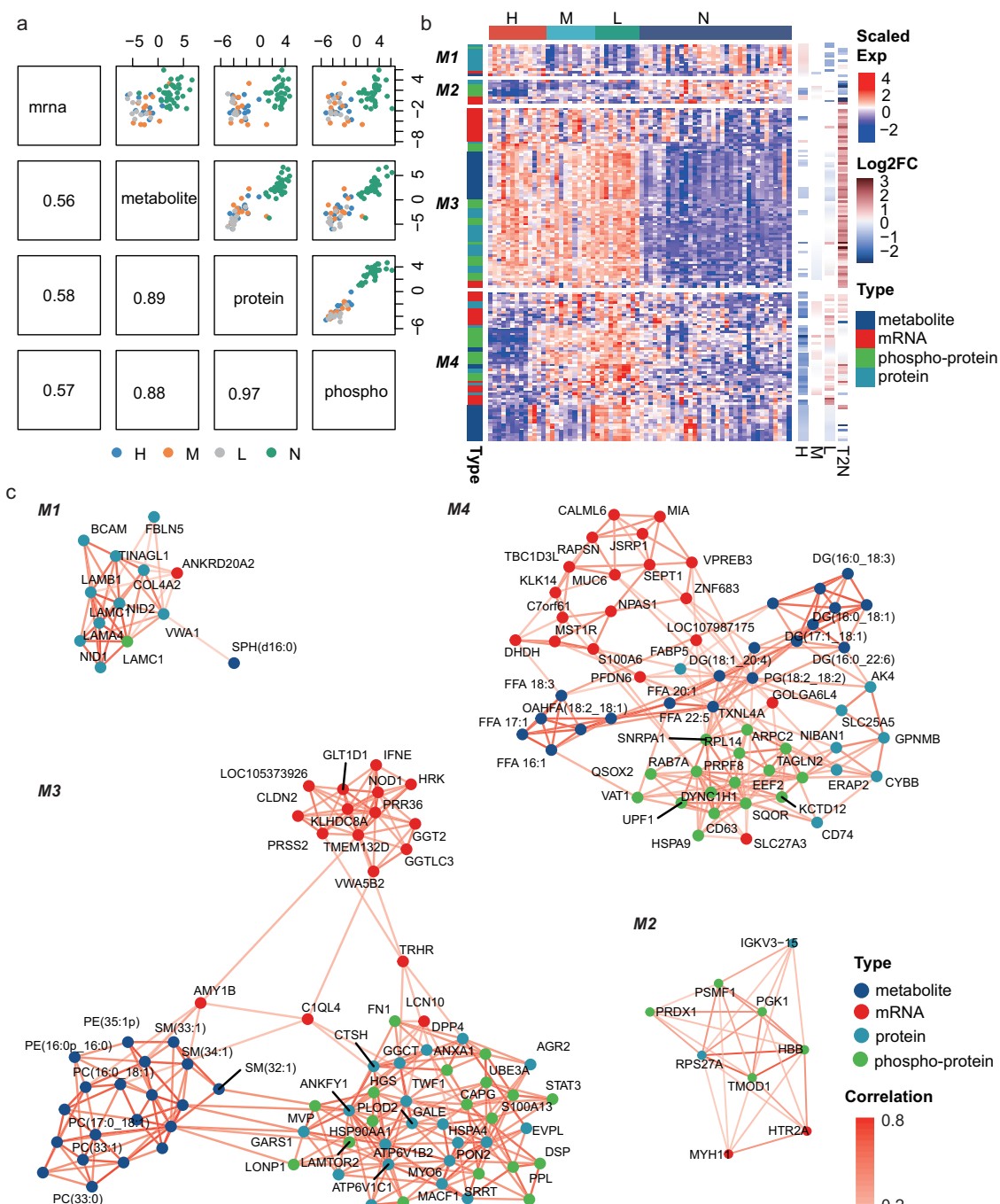

**Fig. 4 | Correlations between different types of omics. a** Correlation between the first DIABLO component determined by different types of omics. **b** Heatmap of the most contributing molecules for the first two components. The four columns on the right showed the log2-transformed fold change (Log2FC) calculated for comparing different groups. H high compared to intermediate and low risks, M intermediate compared to high and low risks, L low compared to high and intermediate risks, T2N tumor compared to paired normal samples. **c** Correlation network for four molecule modules (M1 to M4) based on the multi-omics. Source data are provided as a Source Data file.

implying the different subtypes are oriented from different types of malignant thyrocytes (Fig. 6j).

According to the clinical, molecular and pathway features of the four subtypes, we summarized the four subtypes as low RR and BRAF-like (CS1), high RR and metabolism type (CS2), high RR and immune type (CS3), and high RR and BRAF-like (CS4). Candidate druggable targets for each subtype were identified (Supplementary Fig. S9a, b). These distinctive expression profiles of the four subtypes in the druggable targets, suggesting that different therapeutic strategies

should be applied to different subtypes in PTC (Supplementary Fig. S9b).

**Validation of BRAF-status relevant subtype characters in PTC cells**

The BRAF-status was highly correlated to different subtypes of PTC (Figs. 5a, b and 6g). Meanwhile, for the two subtypes CS2 and CS3 that with fewer BRAF mutations but high recurrence rates, our investigation found that they showed opposed alterations in some

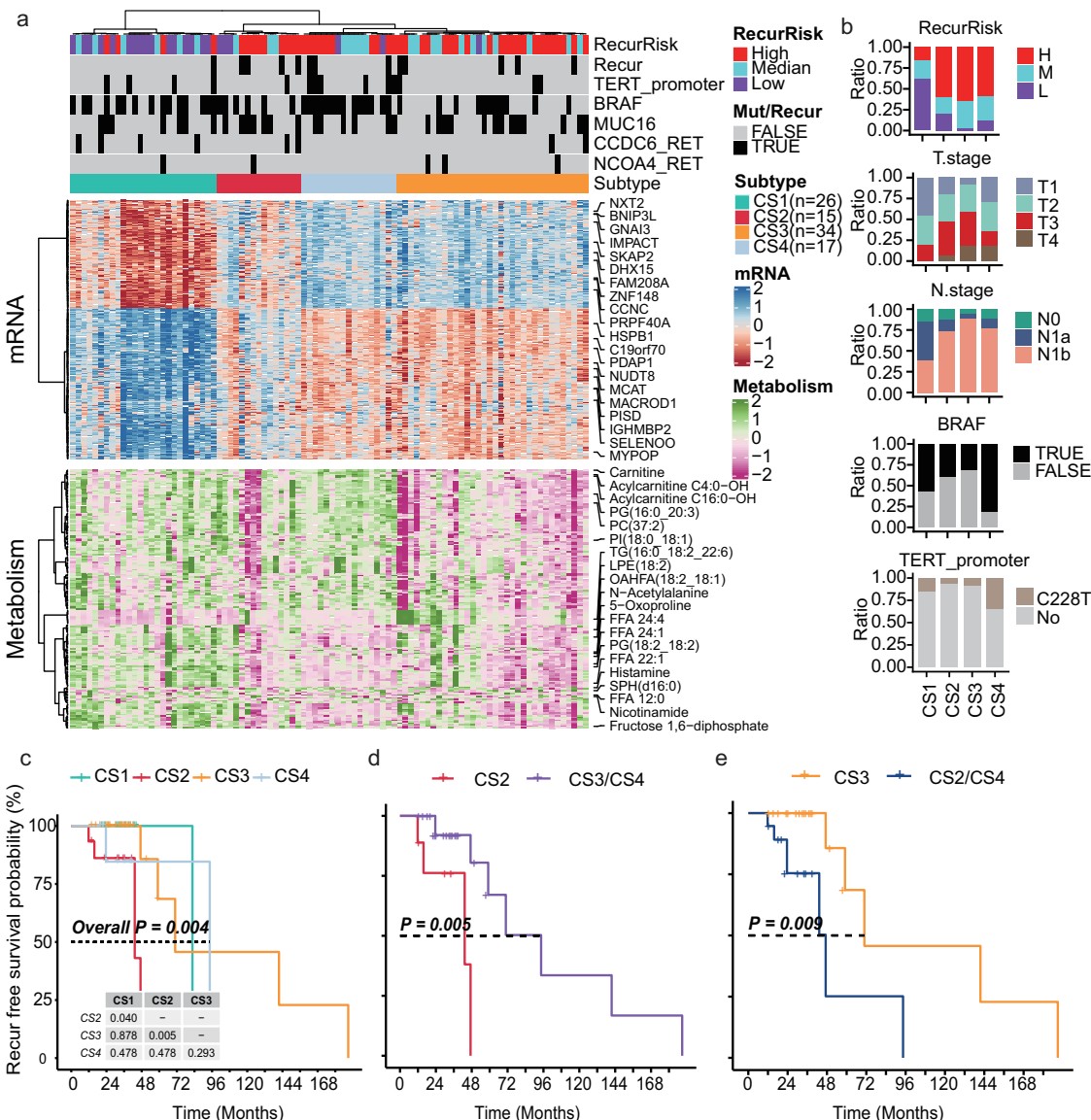

**Fig. 5 | Integrative clustering of the PTC patients based on the transcriptome and metabolomics profile. a** Heatmap showing the clustering results of the PTC patients based on both transcriptomics and metabolomics. **b** Bar plot of the clinical features with significantly different distribution across the four clusters. **c** Kaplan–Meier (KM) plot of the recurrence-free survival probability of the patients in different clusters. P: log-rank test, two-sided. **d** KM-plot of the patients in the cluster CS2 and the other two high-risk enriched subtypes. P: log-rank test, two-sided. **e** KM-plot of the patients in the cluster CS3 and the other two high-risk enriched subtypes. P: log-rank test, two-sided. Source data are provided as a Source Data file.

BRAF-mutation relevant genes, such as LY6K (Fig. 7a and Supplementary Fig. S10a, b). To further validate these correlations, we initially examined the BRAF and its mutant form, BRAFV600E, in different PTC cells (Supplementary Fig. S10c). Notably, IHH4 cells exhibited abundant expression of both BRAF and BRAFV600E in comparison to other PTC cells, and LY6K highly expressed in PTC cells. (Supplementary Fig. S10c, d). To interfere with both BRAF and BRAFV600E expression, we introduced two independent BRAF short-hairpin RNAs (shRNA) into IHH4 cells, resulting in a simultaneous reduction in the expression of BRAF and BRAFV600E, along with a decrease in the downstream factor pMEK1/2 (Fig. 7b and Supplementary Fig. S10e). We observed that LY6K expression decreased alongside BRAF/BRAFV600E down-regulation, indicating a potential reliance on BRAF (Fig. 7c). Further exploration revealed that the restoration of LY6K significantly reversed cell proliferation and tumorigenesis in BRAF knockdown cells (Fig. 7d–f and Supplementary Fig. S10f). These results suggest that the BRAF-status may interact with other factors, such as metabolic

signaling, within PTC, thus cooperatively contributing to the four distinctive subtypes to a certain degree.

Next, to validate the correlation between BRAF-status and metabolic signaling, we conducted metabolomics and transcriptomics analyses using the aforementioned cell lines. Consistent with the observation that multiple metabolites showed increased levels in BRAF-mutant PTC tumor samples (Supplementary Fig. S4b), the controlled IHH4 cells also showed improved levels in a series of metabolites, especially PE and PC species, compared to the BRAF knockdown cells (Supplementary Fig. S10g). The restoration of LY6K lead to significantly decreased levels in multiple metabolites (Fig. 7g) and upregulations of genes in some metabolism pathways like drug metabolism and pentose and glucuronate interconversions (Fig. 7h), in agreement with the reduction of most metabolites and upregulation of genes involved in various metabolism pathways for the LY6K-high subtype CS2 (Fig. 6b, h). Meanwhile, the restoration of LY6K also reversed the pathway impacts generated by BRAF knockdown (Fig. 7h).

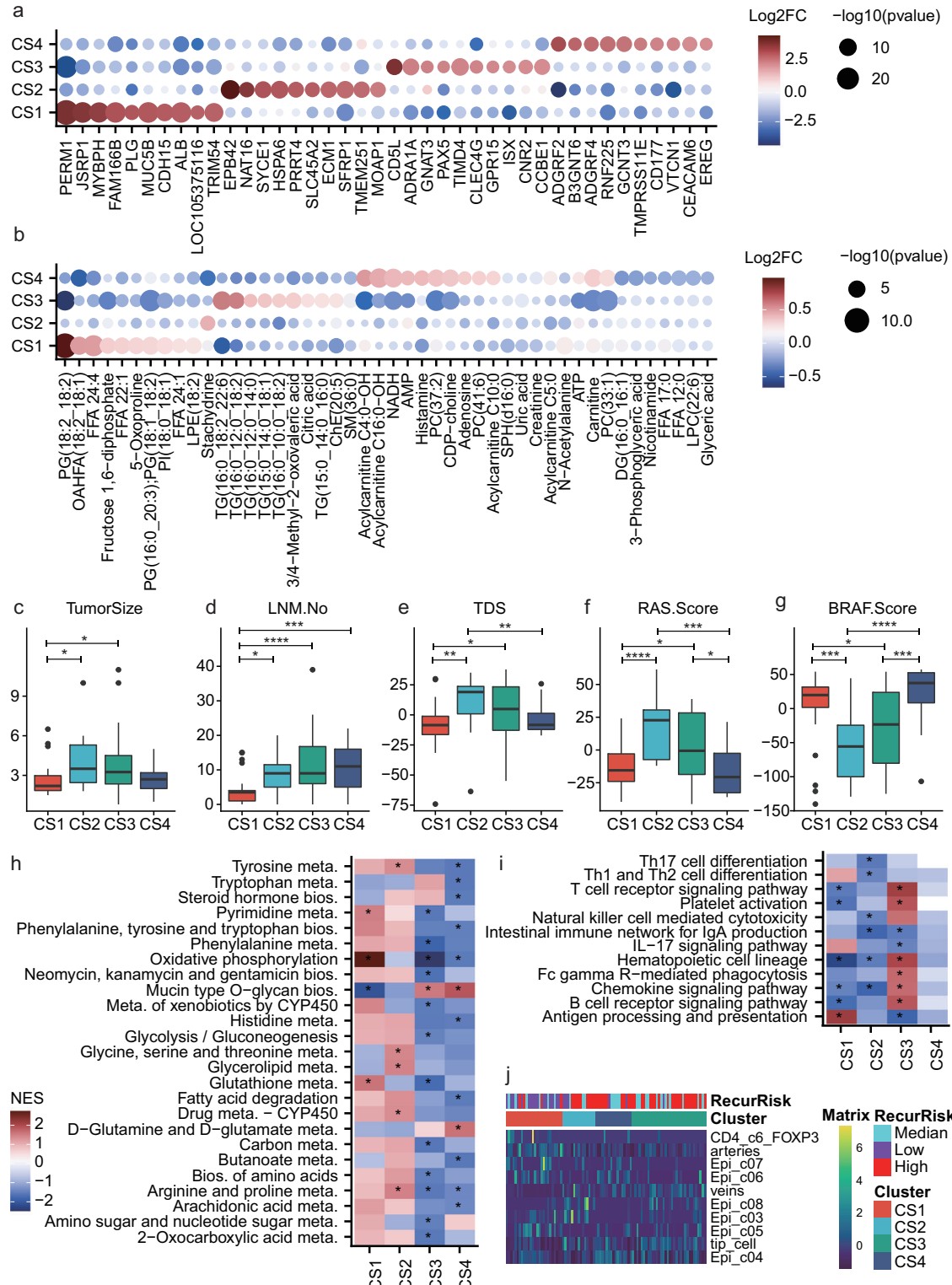

**Fig. 6 | Characterization of the four PTC subtypes. a, b** Significantly upregulated genes (**a**) and metabolites (**b**) in different clusters based on transcriptomics (examined by Deseq2) and metabolomics (examined by Wilcox test, two-sided). Only the top-10 significant items for each subtype are displayed. **c**–**g** Boxplot of tumor size (**c**), number of metastatic lymph nodes (LNM.No) (**d**), tumor difference scores (TDS) (**e**), RAS scores (**f**) and BRAF scores (**g**) across the four clusters (CS1: $n = 26$; CS2: $n = 15$; CS3: $n = 34$; CS4: $n = 17$). T test, two-sided, *$P < 0.05$, **$P < 0.01$, ***$P < 0.001$, ****$P < 0.0001$. In the boxplots, the central line represents median, the

bounds of boxes represent the first and third quartiles, and the upper and lower whiskers extend to the highest or the smallest value within 1.5 interquartile range. **h, i** The metabolism (**h**) and immune pathway (**i**) enrichment results for the four clusters. *$P < 0.05$, by GSEA (Kolmogorov–Smirnov test, one-sided). **j** Heatmap showing different cell compositions of different clusters. Epi epithelium. Only cell types showing significant differences (Kruskal test, two-sided, $P < 0.05$) for at least one subtype is displayed. Source data are provided as a Source Data file.

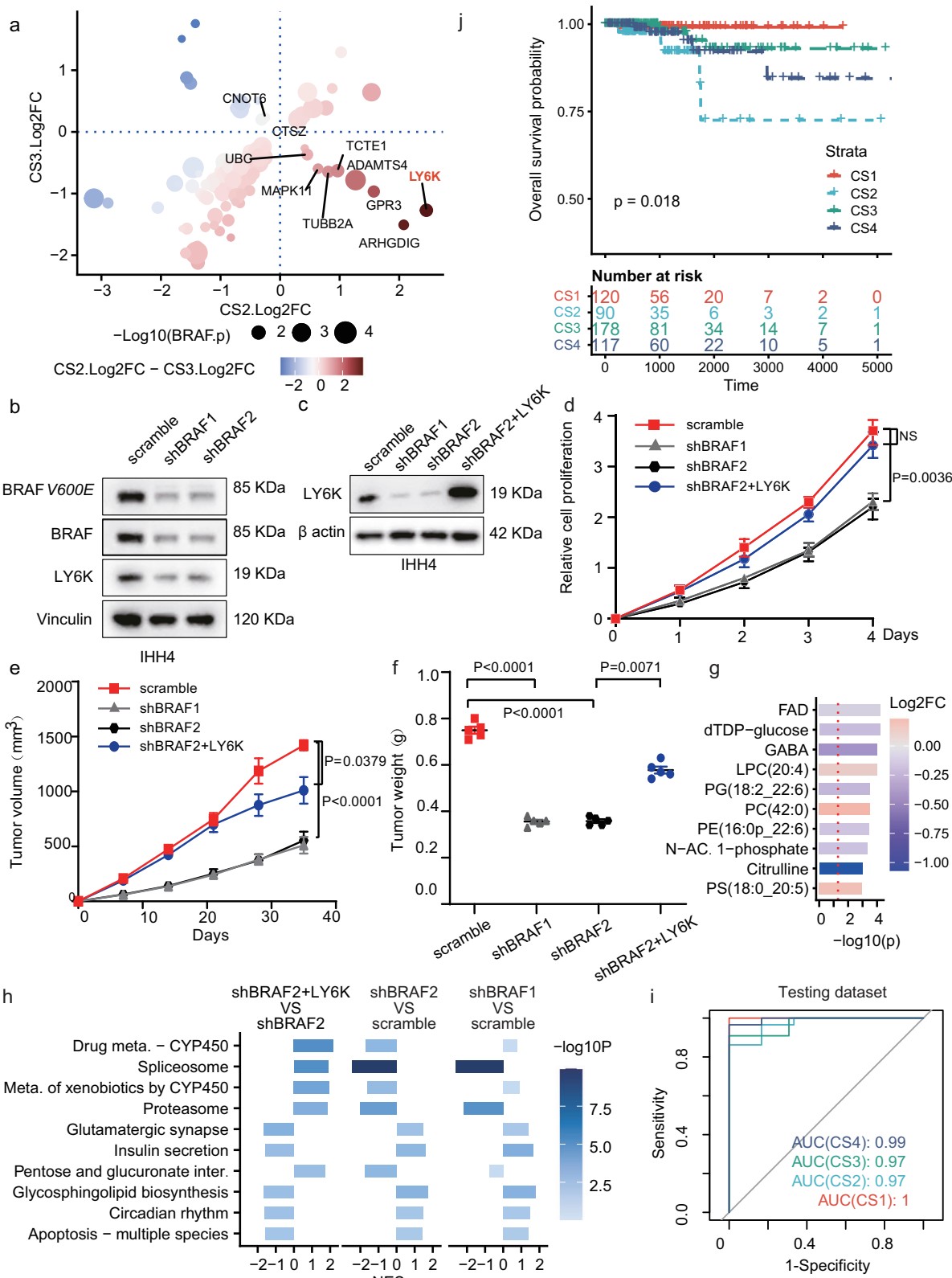

## Validation of the subtypes by machine-learning models

To validate the four PTC subtypes in silico, we split the PTC data into training and testing datasets, and constructed a subtype predictor based on the expression levels of the top-30 ranked genes and metabolites in the training dataset (Supplementary Fig. S10h, i). The predictor can classify the four subtypes accurately in the testing dataset, with the areas under the receiver operator characteristic curves (AUCs)

1, 0.97, 0.97, 0.99 for CS1 to CS4 on the testing dataset (Fig. 7i). Meanwhile, the predicted probabilities of being the subtype CS2 were associated with recurrence-free survival where high CS2 probabilities were associated with unfavorable prognosis (Supplementary Fig. S10j). Moreover, to validate the subtype by external cohorts where only transcriptomics were available, we also trained a subtype predictor based on the top-300 mRNA expressions (mean accuracy across

**Fig. 7 | Validation of the subtypes by both experimental and computational methods. a** Scatter plot of the differential expressions of BRAF-relevant genes in CS2 and CS3 subtypes. The *x* and *y* axes respectively represent the significant Log2FC calculated for comparing CS2 or CS3 to the other clusters, and only genes showed significant differential expression between PTC tumor tissue samples with and without BRAF mutations were considered. Significance: Wilcox test, two-sided, $P < 0.05$. **b** Immunoblotting analysis with indicated antibodies in IHH4 cells transduced with two independent BRAF shRNAs. **c** Immunoblotting analysis of indicated proteins in BRAF knockdown IHH4 and LY6K transduced cells. **d** Proliferation assay of indicated cell lines ($n = 5$). **e, f** Xenograft tumor progression (**e**) and tumor weight (**f**). Mice were injected subcutaneously with IHH4 cells transduced with BRAF shRNA alone or in combination with LY6K ($n = 5$). **g** Bar plot of the top-10

differentially expressed metabolites between shBRAF2 + LY6K and shBRAF2 cells ($n = 5$). *T* test, $P < 0.01$, rank by P. **h** KEGG-based GSEA results for comparing the transcriptomics between different cell lines ($n = 5$). Only pathways with the top-10 NES ($P < 0.05$, Kolmogorov–Smirnov test, one-sided) for comparing shBRAF2 + LY6K VS shBRAF2 are shown. meta. metabolism, inter. interconversions. **i** Receiver operator characteristic (ROC) curve for predictions of CS1 to CS4 subtypes in the testing dataset. **j** KM-plot of the over all survival curves of the four predicted subtypes in the TCGA-PTC dataset. P: Log-rank test. Results (**d–f**) are the mean of biological replicates from a representative experiment, and error bars indicate s.e.m. Statistical significance was determined by *T* test (NS not significant). The experiments were repeated at least three times. Source data are provided as a Source Data file.

tenfold cross-over validations: 0.8234), and utilized it to find the four subtypes in the TCGA-PTC cohort (see "Methods"). Subtypes with similar prognosis patterns can still be recognized (Fig. 7j), where the subtype CS2 was also with the worst prognosis (Supplementary Fig. S10k) as well as fewer BRAF mutations (Supplementary Fig. S10l).

## Discussion

In this study, we performed an integrative investigation of 102 Chinese PTC patients based on multi-omics profiling, including WES, transcriptomics, metabolomics, proteomics and phospho-proteomics. We identified the molecular and pathway characteristics of these Chinese PTC patients from multi-omics perspectives. Consistent with previous reports, common mutations in PTC like BRAF, TERT promoter and gene fusions involved in RET were also revealed. RAS mutations in Chinese PTC cohorts[16,17] were much less than that reported by TCGA[10]. Here, no RAS mutation was identified, probably also due to the geographical as well as sample limitation. The prevalence of TERT C228T promoter mutations in Chinese PTC varied between 4.1 and 9.6% according to previous studies[17,36,37]. The high TERT C228T mutation (14%) frequency here may be related with the large proportion of high RR patients ($n = 48$, 47.06%) in the cohort, since the TERT C228T mutation was found significantly associated with high RR in PTC. In addition, frequent mutations in *MUC16* (36%) were also observed. Although few MUC16 mutation was reported in PTC studies, *MUC16* showed high mutation frequencies and was associated with prognosis in various types of cancers like gastric cancer[38], glioma[39], and melanoma[40], and *MUC16* mutation was found to be associated with better response to immune checkpoint inhibitors in solid tumors[41]. Notably, the discrepancy of the identified mutations compared to TCGA and other cohorts may also probably due to the different mutation filtering strategies. Molecular characters in terms of metabolites, genes, proteins and phosphoproteins were described, these molecular alterations especially focused on the metabolism processes, especially for the glycolysis and pyruvate metabolism. Previous studies have also identified metabolism factors involved in glucose uptake, glycolysis, and lipid metabolism showing significant associations with PTC prognosis[12,13,42] or therapeutic responses[43].

Meanwhile, the molecular differences of PTC with different levels of recurrence risks were also portrayed. Mutations in MUC16, TERT promoter, and various gene fusions were specifically enriched in the high RR PTC patients. The multi-omics-based molecular expressional patterns of different RRs were comprehensively described. It has been reported that elevated post lobectomy serum Tg can be used to predict high recurrence risk[3]. However, the other information about the molecular characteristics of different RRs in PTC was limited. Here, we found the high RR was associated with elevated levels in triglycerides, genes *MMP13* and *CST1*, proteins Tg, PTPRG and VWA1 and phosphorylated EPPK1, ALDH1A1, and LAMC1, etc. Expression of MMP13 was reported to be associated with PTC invasion and metastasis[44]. CST1 upregulation was found to facilitate cell proliferation, motility, epithelial–mesenchymal transition and stemness in PTC[45]. LAMC1 was

reported to show a higher level in samples from PTC patients with metastasis[46]. The specific molecular features of intermediate and low recurrence risks were also described (Fig. 4). The complicated correlations between molecules were also investigated. Interestingly, the high RR showed specific high expressions in the ECM relevant protein dominant correlation network and low expressions in the FFA-centered correlation network. Moreover, the metabolites showed high correlations with proteins and phosphoproteins in general. These findings therefore add to the atlas of biomarkers or targets that may be applied in the diagnosis and treatment of recurrent PTC.

PTC patients were usually stratified into high, intermediate, and low RR according to the ATA recommendations. In this study, we re-stratified the PTC patients into four molecular subtypes based on an integrative clustering analysis using both transcriptomics and metabolomics. The four subtypes were different from the original RR groups, where the first subtype was featured by low RR patients while the other subtypes were mainly composed of the high and intermediate RRs. Biologically, these four subtypes showed remarkable differences in terms of hallmark mutations, PTC-relevant gene and metabolite expressions, epithelial cell compositions, as well as metabolism and immune pathway profiles. Clinically, the subtypes were also enriched by different pathological factors including the disease stages, lymph node metastasis and tumor invasion states. Importantly, the four subtypes showed a significant difference in prognosis, where the subtype CS2 (high RR, less BRAF mutations, upregulation in metabolism) showed the most unfavorable recurrence-free prognosis outcomes, and the subtype CS3 (high RR, less BRAF mutations, upregulations in immune pathways) showed a relatively good prognosis among the three high RR subtypes, highlighting the important roles of metabolism and immune pathways in PTC recurrence. Moreover, the expressional patterns of druggable targets for the four subtypes were distinctive, suggesting subtype-specific treatment may be needed. The redefined subtypes suggest ATA risk stratification should not be used as one single predictor, other molecular profiles should be taken into consideration as well to do better and precision management of PTC recurrences.

Overall, we revealed the molecular basis of PTC with different RRs and proposed an effective molecular stratification strategy. We also illustrated the PTC subtypes relevant molecular characteristics, identified potential drug targets, constructed subtype predictors and highlighted the important role of metabolism in PTC, thus can provide guidance for PTC stratification and promoting precision diagnosis and treatment.

## Methods
### Clinical sample collection

All procedures performed in our study were in accordance with the ethical standards of our institutional research committee and with the 1964 Helsinki declaration and its later amendments or comparable ethical standards. The consecutive samples used for this study were selected from patients diagnosed with PTC from Oct 2014 to Jul 2021 at

Fudan University Shanghai Cancer Center (FUSCC) in China. The sample collection, store, and quality control were in accordance with the standard operation procedures of the Institutional Tissue Bank (ITB) of FUSCC. The samples were detached from the human body and stored in liquid nitrogen within 30 min, and they were made into frozen sections and paraffin-embedded sections at the same time, which were then stained by hematoxylin and eosin. All hematoxylin and eosin slides of the samples were subjected to evaluation for histopathological morphology and tumor components by expert pathologists. The samples enrolled in this study should meet the following criteria: (1) the percentage of tumor cell nuclear (tumor cell nuclear/total cell nuclear) ≥80%, (2) the percentage of total cells ≥80% (cell area/ total tissue section area) and (3) the percentage of necrosis ≤20% (necrotic tissue area/total tissue section area). The clinical information of the enrolled PTC patients were also recorded, including age, gender, sex, tumor size, lymph node metastasis (LNM), extrathyroidal extension (ETE), extranodal extension (ENE), number of metastatic lymph nodes (LNM.No), TNM staging (AJCC cancer staging system 8th edition) and RR stratification (2015 ATA guideline[47]). Each patient provided a written informed consent for his/her specimens and information to be used for research and stored in the hospital database, and this study was approved by the Ethical Committee of the FUSCC.

## DNA library preparation and WES

The exome DNA sequences were enriched from 0.4 μg genomic DNA using Agilent SureSelect Human All Exon V6 kit according to the manufacturer's protocol. DNA fragments were end-repaired and phosphorylated, followed by A-tailing and ligation at the 3' ends with paired-end adaptors. DNA fragments with ligated adapter molecules on both ends were selectively enriched in a PCR reaction. Then, libraries hybridize with liquid phase with biotin-labeled probe, and use magnetic beads with streptomycin to capture the exons of genes. Captured libraries were enriched in a PCR reaction to add index tags to prepare for sequencing. Products were purified using the AMPure XP system (Beckman Coulter, Beverly, USA), DNA concentration was measured by Qubit®3.0 Flurometer (Invitrogen, USA), libraries were analyzed for size distribution by NGS3K/Caliper and quantified by real-time PCR (3 nM). At last, DNA library were sequenced on Illumina for paired-end 150 bp reads. The clustering of the index-coded samples was performed on a cBot Cluster Generation System using Illumina PE Cluster Kit (Illumina, USA) according to the manufacturer's instructions. After that, the DNA libraries were sequenced on the Illumina platform and 150 bp paired-end reads were generated.

## RNA library preparation and RNA-seq for human samples

Briefly, mRNA was extracted and purified from the total RNA of the fresh frozen tissues using poly-T oligo-attached magnetic beads. RNA integrity was measured using the RNA Nano 6000 Assay Kit of the Bioanalyzer 2100 system (Agilent Technologies, CA, USA). Fragmentation was carried out using divalent cations under elevated temperature in the First Strand Synthesis Reaction Buffer (5X). First-strand cDNA was synthesized using random hexamer primer and M-MuLV Reverse Transcriptase (RNase H). Second strand cDNA was synthesized by DNA Polymerase I and RNase H. Remaining overhangs were converted into blunt ends via exonuclease/polymerase activities. After adenylation of 3' ends of DNA fragments, Adaptor with hairpin loop structure were ligated to prepare for hybridization. To select cDNA fragments of preferentially 370–420 bp n length, the library fragments were purified with the AMPure XP system (Beckman Coulter, Beverly, USA). Then PCR was performed with Phusion High-Fidelity DNA polymerase, Universal PCR primers, and Index (X) Primer. At last, PCR products were purified (AMPure XP system) and library quality was assessed on the Agilent Bioanalyzer 2100 system. The library preparations were sequenced on an Illumina Novaseq platform, and 150 bp paired-end reads were generated.

## RNA library preparation and RNA-seq for cell lines

Total RNA was isolated from cells/tissues using the Magzol Reagent (Magen, China) according to the manufacturer's protocol. The quantity and integrity of RNA yield was assessed by using the K5500(Beijing Kiaio, China) and the Agilent 2200 TapeStation (Agilent Technologies, USA) separately. Briefly, the mRNA was enriched by oligodT according to instructions of NEBNext® Poly(A) mRNA Magnetic Isolation Module (NEB, USA). And then fragmented to approximately 200 bp. Subsequently, the RNA fragments were subjected to first-strand and second-strand cDNA synthesis followed by adaptor ligation and enrichment with a low-cycle according to instructions of NEBNext® Ultra™ RNA Library Prep Kit for Illumina. The purified library products were evaluated using the Agilent 2200 TapeStation and Qubit (Thermo Fisher Scientific, USA). The libraries were sequenced by Illumina (Illumina, USA) with paired-end 150 bp at Ribobio Co. Ltd (Ribobio, China).

## Metabolomics profiling

**Sample preparation.** Samples were prepared by extracting metabolites through a chloroform/methanol/water system. In brief, sheared tissues were weighed and then 500 μL methanol with internal standards (including 50 μM ʟ-methionine sulfone and 50 μM ᴅ-camphor-10-sulfonic acid for capillary electrophoresis-mass spectrometry [CE−MS] analysis; carnitine C2:0_d3 at 0.8 μg/ mL, carnitine C8:0_d3 at 0.8 μg/mL, carnitine C16:0_d3 at 0.5 μg/ mL, palmitic acid-d3 at 0.8 μg/mL, ceramide d18:1-d7/18:0 at 0.8 μg/mL, lyso-phospatidylcholine (LPC) 17:0-d5 at 0.8 μg/mL, phosphatidylcholine (PC) 17:0/22:4-d5 at 0.8 μg/mL, and tria-cylglycerol (TAG) 15:0/18:1/15:0-d5 at 0.8 μg/mL for liquid chromatography-mass spectrometry [LC−MS] analysis) were added. Mixed grinding apparatus (Scientz-48) was used for homogenization (35 Hz, 2 min) followed by the addition of 500 μL chloroform and vortex for 30 s. After phase breaking using 200 μL water and centrifugation (13,000 × g, 4 °C, 15 min), the resulting extract was divided into three fractions: one for CE−MS, one for carnitine and acyl-carnitines analysis by LC−MS, and one sample was used for LC−MS-based lipidomics. In total, 300 μL hydrophilic layer was transferred for ultrafiltration through a 5-kDa cutoff filter (Millipore, cat. UFC3LCCNB-HMT). Simultaneously, the quality control sample was prepared by combining the aqueous phase from each sample and then filtered. Then samples were vacuum-dried and stored at −80 °C until CE−MS analysis. For carnitine and acyl-carnitines analysis, 150 μL hydrophilic layer and 100 μL hydrophobic layer were freeze-dried. The quality control sample was also prepared by combining the aqueous phase and then vacuum-dried to evaluate the analytical quality. Overall, 300 μL hydrophobic layer was collected and freeze-dried for lipidomics analysis. At the same time, the quality control sample was prepared by combining the hydrophobic layer from each sample and then vacuum-dried. For cell samples, after gently rinsed in 5% mannitol solution and instantly frozen in liquid nitrogen, cells were lysed with 700 μL methanol containing internal standards and were scraped off from the dish, mixed with 700 μL chloroform. Subsequently, 280 μL water was added. After centrifugation at 13,000 × g for 15 min at 4 °C, 380 μL of aqueous phase was filtrated and freeze-dried for CE−MS analysis and 380 μL hydrophobic layer was collected and freeze-dried for lipidomics analysis. The quality control sample was also prepared and then vacuum-dried.

**Mass spectrometry.** CE−MS analysis was conducted on CE (G7100A, Agilent) couple to time of flight (TOF) mass spectrometry (G6224A, Agilent)[48]. The fused silica capillary (50 μm i.d. × 80 cm, Human Metabolome Technologies (HMT), Japan) was used for sample separation and the temperature of the capillary was at 20 °C. Two analysis modes were performed. For cation mode, a positive voltage

of 27 kV was applied during the CE separation. One M formic acid was used as the background electrolyte and renewed after each ten injections. For TOF/MS, the electrospray (ESI) was performed in the positive ion mode. Parameters were set as follows: nebulizer pressure, 5 psig; dry gas temperature, 300 °C; nitrogen flow, 7 L/min; capillary voltage, 4 kV; fragmentor, 105 V; skimmer, 50 V; Oct RFV, 650 V; acquisition rate, 1.5 spectra/s; mass range, 60–1000 Da. For anion mode, fifty mM ammonium acetate (pH = 8.5) was used as the running electrolyte. During the CE separation, a positive voltage of 30 kV was used. To assist the electroosmotic flow (EOF), an internal pressure of 17 mbar was also applied to the inlet capillary. For TOF/MS, most parameters were identical to those used in the cation mode, except that the electrospray (ESI) was performed in the negative ion mode with a scanning range of 50–1000 Da. Besides, the voltage of the capillary and fragmentor were reset at 3.5 kV and 125 V, respectively. LC–MS analysis was performed by an ACQUITY UPLC system (Waters) coupled with a tripleTOF™ 5600 plus mass spectrometer (AB SCIEX). For acyl-carnitines analysis, the mobile phases consisted of phase A = water + 0.1% formic acid and phase B = acetonitrile + 0.1% formic acid. Lipidomics analysis was conducted through the C8 AQUITY column (2.1 mm × 100 mm × 1.7 μm, Waters, Milford, MA) and liquid chromatography was performed with phase A = 40:60 water: acetonitrile + 10 mM ammonium acetate and phase B = 90:10 2-propanol: acetonitrile + 10 mM ammonium acetate[49]. Briefly, lyophilized samples were reconstituted in chloroform/methanol (2:1, v/v) and diluted threefold in ACN/IPA/H2O (65:30:5, v/v/v/) containing 5 mM ammonium acetate. The flow rate was set as 0.26 mL/min and the column temperature was 55 °C. The elution gradient started at 32% B, was held at this concentration for 1.5 min, was linearly increased to 85% B at 15.5 min, reached 97% B at 15.6 min, and was held at this concentration for 2.4 min. Finally, the column was returned to 32% B within 0.1 min and held at this concentration for 1.9 min for equilibration. The total run time was 20 min. In both ESI (+) and ESI (−) modes, TOF MS full-scan and information-dependent acquisition (IDA) were performed in parallel to acquire high-resolution MS and tandem-MS data simultaneously. In the positive mode, ion source gas 1 and gas 2 were set to 50 psi, curtain gas to 35 psi, temperature to 500 °C, ion spray voltage floating (ISVF) to 5500 V, and collision energy (CE) to 30 V with a collision energy spread (CES) of ±15 V. In the negative mode, ion source gas 1 and gas 2 were set to 55 psi, curtain gas to 35 psi, temperature to 550 °C, ISVF to −4500 V, and CE to −30 V with CES of ±10 V. In the IDA setting, candidate ions with top five intensity were selected and subjected to high-resolution tandem-MS analysis. All samples were randomized with respect to run order to avoid batch effects. In addition, the quality control samples were identically inserted into the analytical sequence to monitor the reproducibility of the analytical method.

**Metabolite identification, quantification, and data normalization.** For CE–MS-based metabolites, the qualitative analysis of metabolites was performed using the pre-analyzed metabolite standard library (HMT), and internal standards were used to adjust the migration time and standardize the metabolite intensity. Peak extraction and identification were carried out with Quantitative Analysis Software (Agilent). Acyl-carnitines identification was based on the mass-to-charge ratio ($m/z$), retention time and MS/MS pattern. Lipid identification was based on exact mass and MS/MS pattern. The applied database search engines were HMDB, Metlin (https://metlin.scripps.edu), and LIPID MAPS. Peakview workstation (AB SCIEX) was used to check MS/MS information of metabolites and Multiquant (AB SCIEX) was used to obtain the peak areas of identified metabolites. The raw data from CE–MS and LC–MS were normalized by corresponding internal standards and tissue weight to minimize errors arising from the sample pretreatment and analysis procedures as much as possible.

## Proteomics and phospho-proteomics profiling

**Sample preparation.** The samples were homogenized in lysis buffer consisting of 2.5% SDS/100 mM Tris-HCl (pH 8.0). Then the samples were subjected to treatment with ultrasonication. After centrifugation, proteins in the supernatant were precipitated by adding four times of pre-cooled acetone. The protein pellet was dissolved in 8 M Urea/100 mM Tris-Cl. After centrifugation, the supernatant was used for the reduction reaction (10 mM DTT, 37 °C for 1 h), followed by an alkylation reaction (40 mM iodoacetamide, room temperature/dark place for 30 min). Protein concentration was measured by the Bradford method. Urea was diluted below 2 M using 100 mM Tris-HCl (pH 8.0). Trypsin was added at a ratio of 1:50 (enzyme: protein, w/w) for overnight digestion at 37 °C. The next day, TFA was used to bring the pH down to 6.0 to end the digestion. After centrifugation (12,000 × $g$, 15 min), the supernatant was subjected to peptide purification using Sep-Pak C18 desalting column. The peptide eluate was vacuum-dried and stored at −20 °C for later use. Phosphopeptide enrichment was performed totally according to a previous study[50].

**LC–MS/MS analysis.** LC–MS/MS data acquisition was carried out on an Orbitrap Exploris 480 mass spectrometer coupled with an Easy-nLC 1200 system. Peptides were loaded through auto-sampler and separated in a C18 analytical column (75 μm × 25 cm, C18, 1.9 μm, 100 Å). Mobile phase A (0.1% formic acid) and mobile phase B (80% ACN, 0.1% formic acid) were used to establish the separation gradient. A constant flow rate was set at 300 nL/min. For DDA mode analysis, each scan cycle consists of one full-scan mass spectrum ($R = 60$ K, AGC = 300%, max IT = 20 ms, scan range = 350–1500 $m/z$) followed by 20 MS/MS events ($R = 15$ K, AGC = 100%, max IT = auto, cycle time = 2 s). HCD collision energy was set to 30. Isolation window for precursor selection was set to 1.6 Da. The former target ion exclusion was set for 35 s.

**Database search.** MS raw data were analyzed with MaxQuant v1.6.6 using the Andromeda database search algorithm. Spectra files were searched against the UniProt Human proteome database using the following parameters: LFQ mode was checked for quantification; Variable modifications, Oxidation (M), Acetyl (Protein N-term) & Deamidation (NQ); Fixed modifications, Carbamidomethyl (C); Digestion, Trypsin/P; The MS1 match tolerance was set as 20 ppm for the first search and 4.5 ppm for the main search; the MS2 tolerance was set as 20 ppm; Match between runs was used for identification transfer. Search results were filtered with 1% FDR at both protein and peptide levels. Proteins denoted as decoy hits, contaminants, or only identified by sites were removed, and the remaining identifications were used for further quantification analysis.

## WES data analysis

Adaptors and low-quality reads of the WES sequencing data were removed by Trimmatic (v 0.39)[51], and the data quality was examined by fastqc (v 0.11.9)[52]. Then, the sequencing data were aligned to the human genome reference (GRCh38/hg38) using BWA (v 0.7.17)[53] and samtools (v 1.8)[54]. The somatic gene mutations of tumor samples with matched normal samples sequenced were called by the function of VarScan2 (v 2.4.4)[55], and the variants of tumor-only samples were called based on the function mpileup2cns of VarScan2 and stringent downstream filters. The called variants were annotated with Annovar[56] (version updated in 2020-06-08) according to multiple databases including refGene, knownGene, Exome Aggregation Consortium (ExAC03), Catalogue Of Somatic Mutations In Cancer (cosmic70), avsnp147, 1000 Genomes Project (2015_08), exome sequencing project (esp6500siv2_all) and clinvar_20220320.

To obtain high-quality somatic variants for the tumor-only samples, stringent downstream filters were used. These filters included a base coverage of a minimum of 200 read depth and variant allele fraction (VAF) of 20 % in tumor, and the variants should be at a

frequency higher than 1% in the 1000 Genomes Project, ESP6500 or ExAC database, or present in the COSMIC with two or more occurrences.

### TERT promoter mutation

The telomerase reverse transcriptase (TERT) promoter mutation (C228T/C250T) is determined using amplification-refractory mutation system quantitative polymerase chain reaction (ARMS-qPCR) as reported in the previous study[57].

### RNA-seq data quantification

RNA-seq reads were aligned to the genome reference (GRCh38/hg38 for human samples and mm10 for cell lines) using HISAT2 (v2.0.5)[58]. HTseq (v 2.0.2)[59] was utilized to count the read numbers of each gene. Normalized gene expression matrix was obtained based on the counts function of DESeq2[19] (v 1.26.0) with parameter normalized = TRUE.

### Gene fusion

Gene fusions were detected from the RNA-seq data using arriba (v 2.3.0)[60] and STAR-fusion (v 1.4.0, https://github.com/STAR-Fusion/STAR-Fusion/wiki). The results from the two methods were further annotated and filtered base on annoFuseData (https://github.com/d3b-center/annoFuseData) and annoFuse (V 0.90.0)[61], and only fusions with JunctionReadCount >3 and evaluated as high or median confidence were retained.

### Differential expression analysis

For the RNA-seq data, DESeq2[19] (v 1.26.0) was applied to find the differentially expressed genes. For the metabolism/proteomics/phospho-proteomics data of PTC patient samples, the differential expressions of each molecule was examined by Wilcox test (paired, two-sided). Log2FCs between the samples were also calculated. $P$ values were adjusted by the Benjamini– Hochberg method.

### Multi-omics characterization of PTCs with different RRs

For each RR type, we recognized the RR type associated molecules by comparing the expressions of molecules (metabolites, mRNAs, proteins, phosph-proteins) between this RR type and the other two RR types using Wilcox test (unpaired, two sides) and corresponding Log2FCs were calculated.

### Integrative correlation analysis

The DIABLO[26] method (mixOmics R package, v 6.10.9) was applied to the four types of omics data (transcriptomics, metabolomics, proteomics, phosph-proteomics, and only molecules showed significant differences between tumor and normal tissues or specific type of RR were taken into account), with the samples covered by all four types of omics and labeled as high RR, intermediate RR, low RR and normal tissue. The DIABLO method aims to obtain the common information across multi-omics data by selecting a subset of molecules which not only maximize the inter-correlations among omics but also discriminate between different phenotypic labels. The expressional matrixes and the labels were taken as the input of DIABLO, the latent component number was set as 3, and the number of representative molecules to select for each latent component considering each type of input omics data was set as 20.

### Integrative clustering of PTC patients

First, we tried to cluster the PTC patients based on both transcriptomics and metabolomics profiles. Here, ten different multi-omics clustering methods, including SNF, PINSPlus, NEMO, COCA, LRAcluster, ConsensusClustering, CIMLR, MoCluster, iClusterBayes, IntNMF were performed on our data using the MOVICS[62] R package (v 0.99.17). These methods generated ten clustering records, and a similarity matrix describing to what extant different samples were grouped into the same clusters in terms of the ten clustering records was obtained. Then, a hierarchical clustering algorithm was applied on the consensus similarity matrix, and the number of clusters was set as 4 (to ensure each cluster with was with more than ten samples).

### TDS, BRAF scores, and RAS scores

We calculated the mean log2-transformed expression levels of 16 thyroid function-relevant genes defined by the TCGA-PTC study[34] as the TDS scores. Similarly, the BRAF and RAS scores were calculated based on the mean expression of the upregulated signature genes in the BRAFV600E-mutated and RAS-mutated samples from the TCGA-PTC mRNA expressions.

### The scRNA-seq-based prediction of cell compositions in bulk samples

The single-cell RNA-seq data of PTC samples as well as the annotated cell types were utilized as the input of dampened weighted least squares (DWLS R package, v 0.1.0, https://CRAN.R-project.org/package=DWLS) to train a cell decovolution model. Then, the trained DWLS model was applied on the transcriptomics data of the PTC bulk tissue samples to estimate the potential cell compositions of each tissue sample, and a cell-type composition matrix was obtained.

### Identification of druggable targets

For each subtype, the subtype-associated genes and metabolites were recognized by examining the differential expressions between one subtype and the other three subtypes using DESeq2 for transcriptomics and limma for metabolomics. Then, the top-10 significant and specifically highly expressed genes for each subtype was recognized using the runMarker function of the MOVICS package. Then, druggable targets among the top-ranked subtype-associated genes were selected based on the DGIdb database[63].

### Subtype prediction

The transcriptomics and metabolomics data of the 97 PTCs were partitioned into training (60%) and testing datasets (40%). The importance of the metabolites and genes in predicting the subtypes were estimated based on random forest (RF)[64] method using the randomForest R package (v 4.6-14). Then, a subtype predictor was trained based on the expressional profiles of the top-30 metabolites and top-30 genes using the RF method. The model was applied on the testing dataset, and the prediction performance was evaluated by ROC curves.

Since the TCGA-PTC cohort only had the transcriptomics data. We trained another subtype predictor based only the transcriptomics data. The importance of the genes in predicting the subtypes were also estimated based on RF method. The subtype predictor was trained based on the expressional profiles of the intersection of the top-300 genes in our data set and the genes covered by the TCGA-PTC transcriptomics data using the linear discriminant analysis. The model was applied on the TCGA-PTC cohort to predict the subtype for each sample.

### Cell culture, shRNA transduction and cell growth measurement

Human IHH4 (sex: male) was purchased from JCRB Cell Bank (Catalog: JCRB1079). Human TPC1 (sex: female) was purchased from Sigma-Aldrich (Catalog: SCC147). Human BCPAP(sex: female) was purchased from Cell Bank (Chinese Academy of Sciences, Catalog: SCSP-543). Human HEK293T (sex: female) was purchased from ATCC (ATCC Number: CRL-3216). IHH4, TPC1, BCPAP, and HEK293T cells were cultured in DMEM with 10% FBS. Lentiviral shRNAs were cloned into pLKO.1, the targeted sequences were: shBRAF1- GTTACCTGGCTCAC-TAACTAA; shBRAF2-GAACATATAGAGGCCCTATTG. HEK293T cells were transfected using polyethylenimine (PEI). The transfection mass ratio of plasmids to PEI was 1:3. Lentivirus production was performed using two systems. For the two-plasmid packaging system, psPAX2 and

pVSVg, together with targeting plasmids were co-transfected into HEK293T cells for 48 h to harvest the supernatant of lentivirus. IHH4 cells were incubated with the medium mixed with the indicated supernatant of lentivirus for 24 h. Next, the cells were kept in the normal culture medium and used for further treatment.

Cell proliferation was performed using cell counting Kit 8 according to the manufacturer's requirement. Briefly, indicated cells (1000–2000 per well) were seeded in a 96-well plate. At different time points (24 h for a time point, a total 5 time points), cells were incubated with 100 µl relative culture medium containing 10% CCK-8 assay solution. After 2 h of incubation, plates were measured at 450 nm using a plate reader (Cytation5, Biotech).

## In vivo animal studies

Male BALB/c nude mice (6–8 weeks old) were obtained from the China Medical University and maintained under specific pathogen-free (SPF) conditions. For tumor growth assay, IHH4 cells ($2 \times 10^6$ per mice) in 100 µl of cells suspension (mixed with Matrigel at a 1:1 ratio) were injected subcutaneously. All experiments were carried out according to the regulations set by the Ethics Committee of China Medical University. All animal experiments were performed in accordance with a protocol approved by the Institutional Animal Care and Treatment in Biomedical Research of China Medical University. When used in a power calculation, our sample size predetermination experiments indicate that 5 mice per group can identify the tumor size and weight ($P < 0.05$ with 100% power). Animals were randomly assigned to different groups. Six- to 8-week-old male nude mice were used for subcutaneous injection of human thyroid cancer cells. Tumor cells in 30 µl of growth medium (mixed with Matrigel at a 1:1 ratio) were injected subcutaneously using a 100-µl Hamilton Microliter syringe. Tumor size was measured once a week using a caliper, and tumor volume was calculated using the standard formula $0.5 \times L \times W2$, where L is the longest diameter and W is the shortest diameter. The maximal tumor burden permitted by the ethics committee is no more than 1500 mm³. When the tumor burden reached 1500 mm³, mice were euthanized, and the tumors were dissected for further analysis. All the tumors were removed, photographed and weighed.

## Immunoblotting

Cells were harvested and extracted proteins using ice-cold lysis buffer (150 mM NaCl, 1% Triton X-100, 1 mM EDTA, 1 mM EGTA, 2.5 mM sodium pyrophosphate, 1 mM β-glycerolphosphate, 20 mM Tris-HCl, pH 7.5, with protease inhibitor cocktail). After denaturation, samples were subjected to SDS-PAGE electrophoresis and immune-blotting assay. Primary antibodies against BRAFV600E was purchased from Abcam, Inc. (Cam, UK) (Catalog: ab228461, dilution 1:500), and BRAF was from Santa Cruz Biotechnology, Inc. (DAL, USA) (Catalog: sc-5284, dilution 1:1000). Primary antibodies against LY6K were purchased from Beyotime Biotechnology, Inc. (SH, CN) (Catalog: AG5061, dilution 1:1000), and Actin was purchased from Proteintech Group, Inc. (IL, USA) (Catalog: 81115-1-RR, dilution 1:2000). Vinculin was purchased from Santa Cruz Biotechnology, Inc. (Texas, USA) (Catalog:sc-73614, dilution 1:2000). Primary antibodies pMEK1/2 (ser 217/221) and MEK1/2 were purchased from Cell Signaling Technology, Inc. (MA, USA) (Catalog: 8727T, dilution 1:2000 and Catalog: 9154T, dilution 1:1000).

## Statistical analysis

Detailed computational and statistical methods are reported in "Methods" or figure legends. All statistical analyses were performed by R (v 3.6.3 and v 4.0.4). Survival analysis were performed by survival (v 3.2-7) and survminer (v0.4.9) packages. Pathway enrichment analyses were performed by clusterProfiler[65] (v 3.18.1) package. The statistical tests were two-sided and unpaired by default, and one-sided or paired tests were specifically stated.

## Reporting summary

Further information on research design is available in the Nature Portfolio Reporting Summary linked to this article.

## Data availability

The raw WES and RNA-seq data of the PTC samples have been deposited in the Genome Sequence Archive[66] in National Genomics Data Center[67], China National Center for Bioinformation/Beijing Institute of Genomics, Chinese Academy of Sciences (GSA-Human) under accession code HRA005293 and HRA005382. The raw WES and RNA-seq data are available under restricted access for research purposes only, access can be obtained by the DAC (Data Access Committees) of the GSA-human database. According to the guidelines of GSA-human, all non-profit researchers can obtain access to the data, and the principal investigator of any research group is allowed to apply the data. The access authority can be obtained for Research Use Only. The user can also contact the corresponding author directly. Once access has been approved, the data will be available to download for 2 months. The mass spectrometry proteomics and phospho-proteomics data have been deposited to the ProteomeXchange Consortium via the PRIDE[68] with the dataset identifier PXD044900 and PXD045017. The metabolomics data have been deposited to MetaboLights[69] [www.ebi.ac.uk/metabolights/MTBLS3339]. Transcriptomics and survival data of TCGA-PTC samples were obtained from Genomic Data Commons [https://portal.gdc.cancer.gov/projects/TCGA-THCA]. The scRNA-seq data used in this study are available in the Gene Expression Omnibus repository under accession code GSE184362. Source data are provided with this paper.

## Code availability

The codes are available at https://github.com/diChen310/PTC_multi_omics.

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

## Acknowledgements

This study is supported by the National Natural Science Foundation of China Grants (No. 81972625 to H.-l.P., No. 82072951 to Y.W., No. 82174133 to S.J., No. 82203052 to B.M., No. 82373315 to N.Q.), Liaoning Revitalization Talents Program (XLYC2002035 to H-l.P.), Liaoning Science and Technology Innovation Funding (20230101-JH2/1013 to H.-l.P.), Plan Foundation Project of Education Bureau of Liaoning Province (JYTMS2023580 to H.H.), Science and Technology Innovation Fund (Youth Science and Technology Star) of Dalian (No. 2021RQ009 to D.C.), Innovation program of science and research from the DICP, CAS (DICP I202129 to D.C., DICP I202209 to H.-l.P.), the Program of DMU-1&DICP (UN202206 to S.J.), the Science and Technology Commission of Shanghai Municipality (22Y21900100 to Y.W.), the Shanghai Anticancer Association (SACA-AX202213 to Y.W.). We genuinely appreciate for our colleagues from the Department of Pathology and the Institutional Tissue Bank (ITB) at FUSCC. The authors are also grateful to those who assisted in proteomic analysis at Wuhan Metware Biotechnology Co., Ltd. In particular, the authors would like to express sincere thanks to Xi Zhan, for the help in raw data processing.

## Author contributions

Yu W., Q.J., H.H., H.-l.P. and R.S. conceived the project, H.-l.P. and R.S. supervised the project. N.Q., D.C., B.M., H.H., H.-l.P. and R.S. designed the project and performed most of the experiments, D.C. performed the computational data analysis. N.Q., B.M., Yuting W., Z.N., T.L., J.X. and Yulong W. collected clinical samples and the relevant clinical information. L.Z., H.W., S.J., D.X. and W. Wu helped to analyze the results from a clinical perspective. W. Wang conducted the metabolism profiling experiments. Q.W. and H.H. conducted the in vitro and in vivo experiments for validation during the revision stage. N.Q., D.C., B.M. and H.-l.P. wrote the manuscript with input from all other authors.

## Competing interests

The authors declare no competing interests.
