## [Peer Review File · Nature Communications]

REVIEWER COMMENTS

Reviewer #1 (Remarks to the Author): Expert in PTC proteogenomics and multi-omics

The authors have performed integrated multiomics analysis of PTC patients with different recurrence risks and identified interesting signatures, novel mutations and fusions. The manuscript is interesting for researchers interested in PTC. My enthusiasm for publication here is mitigated by two main issues 1) lack of validation and functional characterization 2) functional characterization of identified mutations and driver oncogenes in the control of the metabolomic signatures. There are no mechanistic studies linking the casual relationship between the identified factors and the pathophysiology of PTC.

Major concerns

- 1) How is the tissue is collected? Do the authors control warm and cold ischemia ? No details on the SOPs employed for the collection, processing and biobanking of tissue is provided. I don't find details on the QC protocol for tissue – tumour content and biomarkers employed for the authentication of tissue samples
- 2) It would be nice to have a table mentioning the range of age, gender, tumor stage, additional therapies or comorbidities for better overview.
- 3) For figure 1, Here you plot tumor stages for specific target genes. Did you check for pathways or genes that are expressed differently in early and later tumor stages?
- 4) Figure 2: There is a low overlap between the transcriptomic with proteomic pathway: did you check for protein modifying enzymes or regulatory pathways responsible?
- 5) Fig 2: Could you identify mutations in any enzymes related to the metabolic pathways in tumor samples?
- 6) The authors claim that some of these metabolite levels has been confirmed in the serum levels of the patients: could you please provide the data for their measurement in the serum?

Minor

1. Legend of Fig S1: basically the whole legend is wrong:the legend for S1a does not match the graph, I think the legend is for S1b and they just *forgot S1a?
- 2 . in S1b legend, there is a typo
3. n S1c legend "comparision"
4. Fig 1a: They should organize the legend in the graph in a better way. It is kind of all over the place, just put it in the same order as the graph above.
5. Fig 1b: legend for the * is missing. $p < 0.05$?
6. Fig 1 c&d: Sometimes they annotate recurrence risk as "RR" and sometimes as "RecurRisk", they should decide on one
7. Row 101-106 in the text, they talk about mutations enriched in the high RR patients and cite Fig 2b and 2d but those figures show differentially expressed molecules (DEMs). Am I missing something here?
8. Fig S4 only has legends for 2 of the 5 letters in the graph
9. Row 225-228 in the text makes no sense? something is missing

10. In Fig S5b remove one "the".
11. Fig S6 legend a: differet was written instead of different
12. Fig S7 legend a: more description of the analysis is needed, was this done from the Drug Gene Interaction Database?
13. The supplemental Figure 7 (S7) was not cited at all in the paper.

Reviewer #2 (Remarks to the Author): Expert in thyroid cancer metabolism

The work of Ning Qu and colleagues is based on an extensive molecular profiling (multi-omics integrated analysis) of papillary thyroid carcinoma samples (PTCs) displaying – following ATA guidelines – different recurrence risks (RR). The work identified distinctive and relevant molecular features of PTCs, allowing to redefine the 3 RRs categories (low, middle, high risk) into four distinct molecular subtypes based on genomic, transcriptional, proteomic and metabolic data. The work is of high relevance to the field and it also represents a potential guideline for a better characterization of RR in other tumors. The paper is well-written, and the methods reported herein are well described, allowing to reproduce the study.

Overall, the results of the work support the conclusions even though, as detailed in the comments below, I have some concerns and suggestions to hopefully improve the overall quality of the manuscript.

Major:

- 1) In the genomic analysis of somatic mutations in PTC samples, the authors do not identify mutations in RAS genes, considering that in the TCGA collection, as well as in many other large or small cohorts (GENIE v3 among them), the cumulative mutation frequency for RAS genes (H-, N- and K-RAS) is even 10-15%. Can this be explained only by geographical reasons? How the authors explain this discrepancy? Did the authors check – by standard Sanger sequencing – some of the PTC samples that were negative for most of the known mutations (at least BRAF and RET/PTC fusions)?
- 2) The frequency of TERT promoter's mutation is highly variable in PTC. It ranges from about 30%, as observed by the GENIE consortium, to very low frequencies such as in TCGA and in a work from Nikiforov's group (Panbianco et al., 2019) that found this mutation in 7% of PTC cases. As this mutation does not fall into the WES panel, the authors used ARMS-qPCR to detect it. Did the authors validate (here or in the previous work) the accuracy of this detection system? Is there the possibility that the measured percentage is biased by the detection method, or it reflects the mutations frequency in Chinese PTCs? Are there other works on similar (by geographical distribution) populations regarding PTCs? If so, please include in the manuscript and discuss on it.
- 3) Considering the relevant number of samples analyzed by transcriptome analysis, did the authors identify completely new gene rearrangements (searching for fusion/chimeric transcripts)? If so, please describe them and also verify the expression of related parent genes in the same data sets, possibly commenting on the findings in terms of oncogenic/tumor-suppressive potential of the new chimeric

transcripts.

4) In the integrated analysis the authors focus on common pathways affected. How do the authors explain the low overlap - in terms of enriched pathways - among transcriptome, proteome and phosphoproteome? Is it due to the low correlation between single omic data sets (transcriptome vs proteome, proteome vs phosphoproteome etc)? If so, could it be relevant to consider also the "application-specific" ones? Could this discrepancy reflect an unknown biological/pathological factor rather than only depend on "technical" variability?

5) When describing the metabolic alterations that the authors identified in PTC (lines 168-172), other - and more relevant than ref. 9 (line 169) - citations could be added to further highlight that multiple independent studies have already identified gene/protein signatures and metabolic changes in glucose uptake, glycolysis and lipid metabolism in PTCs (Chai et al. Surgery 2017; Ma et al. J Clin Endocrinol Metab 2019; Ban EJ et al. Endocrinol Metab 2021; Xu F et al. Front Oncol 2021; Wen et al. Acta Biochim Biophys Sin 2021; Aprile et al., Br J Can 2023). Noteworthy, some of these recent works reported a correlation with poor prognosis, tumor relapse and/or low therapeutic response. These works may be also briefly commented in the Discussion (around line 399) to foster an open field represented by the targeting of metabolic pathways, also in the PTC context.

6) Considering that transcriptome and proteome data indicate immune mediators and immunity-related pathways/processes as differentially expressed, and the omic analyses have been done on bulk tissue biopsies, could the different cell composition - reported on lines 335-338 - depend on the heterogeneity of tumor biopsies?

7) The authors identified by multi-omic integrated cluster analysis the presence of 2 BRAF-like subgroups having completely opposite RR (ie. low and high). Considering the heterogeneous landscape of somatic alterations in the BRAF-mutated PTCs, did the authors check if CS4 cluster is enriched in other (additional) somatic mutations (TERT or other MAPK or PI3K genes) and/or other gene fusions that may account for the completely different RR?

8) Considering the low correlation, as measured by the authors, between transcriptome and proteome/phosphoproteome data in these PTC samples, and considering also that proteins, rather than transcripts, are the bona fide targets of drugs, it would be expected to build a network using proteome/phosphoproteome data rather than mRNA expression. Could the authors comment on this and adjust their analysis making it more focused on deregulated proteins/enzymes?

9) At the end of Discussion the authors comment on the most relevant result of their work, ie. the conclusion that ATA stratification may not be taken as the only single predictor for PTC. It is supported by the findings of the authors. However, how do the authors propose to employ their subclassification based on multiple omics and bioinformatics in a routine clinical practice that is based on the management of hundreds/thousands of PTC patients in clinics or hospitals? Did the authors propose a minimal subtype-specific "signature" of molecules to be tested (eg. a panel made by a specific combination of x genes, y proteins and/or z metabolites etc.) that can be eventually used in the clinical practice with reasonable low costs and in a reasonable amount of time? It would provide a significant

clinical impact and would strengthen the current work.

Minor:

1) The discrepancy of the results on the genomic analysis (compared to TCGA or other large consortia) regarding MUC16 may be due to the filtering process adopted by genomic centers, as some of these genes (including also RNF213) are often considered frequently-mutated “not-specific” cancer genes.

2) In the paragraph “Multi-omics based comparison of tumor and normal tissues of PTC patients”, I suggest removing the sentences with "etc." (lines 125-7) describing the genes/proteins differentially expressed. These sentences, in my opinion, do not add to the manuscript.

3) Lines 129-130 Fibronectin 1 is repeated two times.

4) Lines 190-192. The authors found PTC tumor samples having lower Tg compared to normal samples, but “high RR” PTC samples behave the opposite compared to other tumor subtypes. What could be the explanation for this?

Reviewer #3 (Remarks to the Author): Expert in MS-based proteomics, phosphoproteomics, and metabolomics

Although thyroid cancer is considered a rare cancer that affects the thyroid gland, the estimated global incidence rate is rising quickly. Age, gender, and radiation exposure are a few variables that may increase the risk of thyroid cancer. Recurrence risks (RRs) is categorized into three levels risk stratification: low, intermediate, and high. However, the molecular underpinnings of the various RRs remain poorly known. The identification of biomolecules that may contribute to a better PTC risk stratification has utmost importance for the development of effective diagnostic and treatment strategies. Within this context, the topic of this study is timely and pertinent. Specifically, Qu et al performed a integrated and comprehensive multiomics analysis to identify molecular determinants associated with the four different type of PCT RRs cancer. Overall the text is well structured and the data gathered is sound and support most of the claims made throughout the text. However this reviewer has major points that need to be address by the authors.

1) The authors should state clearly the study's calculating power because 102 samples seemed insufficient for this type of research. This brings me to my second point 2). Although the authors conducted a thorough integrated multiomics analysis, the work is primarily descriptive and to some extent lacks validation or verification. The authors should considered targeted approaches (metabolomics/proteomics) to confirm potential biomarkers/druggable targets claimed to be associated with the various types of PCT RRs cancer . Such a strategy might strengthen the study's conclusions and minimize the impacts of the small sample size.

3) Phopshoproteomics results should clearly indicate the p-site (S/T/Y) identified. Was there an attempt to calculate the occupancy of the p-sites that were identified.

4) Data accessibility: This reviewer was unable to find any indication of data being made available in open repositories like Metabolomic Workbench or PRIDE for proteomics data. This should have been completed and made accessible to reviewers even before submission.

5) A figure demonstrating the experimental design and providing a detailed description of the cohort under study would also be useful. the sampling (tissue), sample extraction, and downstream analysis.

REVIEWER COMMENTS

Reviewer #1 (Remarks to the Author): Expert in PTC proteogenomics and multi-omics

The authors have performed integrated multiomics analysis of PTC patients with different recurrence risks and identified interesting signatures, novel mutations and fusions. The manuscript is interesting for researchers interested in PTC. There are no mechanistic studies linking the casual relationship between the identified factors and the pathophysiology of PTC.

Re: Thanks a lot for the reviewer's interest and suggestions in our research.

We have performed experimental and computational validations about the main findings, as added in Page 27-31, as below:

Validation of BRAF-status relevant subtype characters in PTC cells

The BRAF-status was highly correlated to different subtypes of PTC (Fig 5a-b and Fig 6g). Meanwhile, for the two subtypes CS2 and CS3 that with fewer BRAF mutations but high recurrence rates, our investigation found that they showed opposed alterations in some BRAF-mutation relevant genes, such as LY6K (Fig 7a, Fig S10a-b). To further validate these correlations, we initially examined the BRAF and its mutant form, BRAFV600E, in different PTC cells (Fig S10a). Notably, IHH4 cells exhibited abundant expression of both BRAF and BRAFV600E in comparison to other PTC cells (Fig S10c). To interfere with both BRAF and BRAFV600E expression, we introduced two independent BRAF short-hairpin RNAs (shRNA) into IHH4 cells, resulting in a simultaneous reduction in the expression of BRAF and BRAFV600E (Fig 7b). Then, we investigated the expression of LY6K (Fig S10d). We observed that LY6K expression decreased alongside BRAF/BRAFV600E downregulation, indicating a potential reliance on BRAF (Fig 7c). Further exploration revealed that the restoration of LY6K significantly reversed cell proliferation and tumorigenesis in BRAF knockdown cells (Fig 7d-f and S10e). These results suggest that the BRAF-status may interact with other factors, such as metabolic signaling, within PTC, thus cooperatively contributing to the four distinctive subtypes to a certain degree.

Next, to validate the correlation between BRAF-status and metabolic signaling, we conducted metabolomics and transcriptomics analyses using the aforementioned cell lines. Consistent with the observation that multiple metabolites showed increased levels in BRAF-mutant PTC tumor samples (Fig S4b), the controlled IHH4 cells also showed improved levels in a series of metabolites, especially PE and PC species, comparing to the BRAF knockdown cells (Fig S10f). The restoration of LY6K lead to significantly decreased levels in multiple metabolites (Fig 7g) and up-regulations of genes in some metabolism pathways like drug metabolism and pentose and glucuronate interconversions (Fig 7h), in agreement with the reduction of most metabolites and up-regulation of genes involved in various metabolism pathways for the LY6K-high subtype CS2 (Fig 6b, Fig 6h). Meanwhile, the restoration of LY6K also reversed the pathway impacts generated by BRAF knockdown (Fig 7h).

Validation of the subtypes by machine learning models

To validate the four PTC subtypes in silico, we splited the PTC data into training and testing datasets, and constructed a subtype predictor based on the expression levels of the top-30 ranked genes and metabolites in the training dataset (Fig S10g-h). The predictor can classify the four subtypes

accurately in the testing dataset, with the areas under the receiver operator characteristic curves (AUCs) 1, 0.97, 0.97, 0.99 for CS1 to CS4 on the testing dataset (Fig 7i). Meanwhile, the predicted probabilities of being the subtype CS2 were associated with recurrence-free survival where high CS2 probabilities were associated with unfavourable prognosis (Fig S10i). Moreover, to validate the subtype by external cohorts where only transcriptomics were available, we also trained a subtype predictor based on the top-300 mRNA expressions (mean accuracy across 10-fold cross-over validations: 0.8234), and utilized it to find the four subtypes in the TCGA-PTC cohort (see Methods). Subtypes with similar prognosis patterns can still be recognized (Fig 7j), where the subtype CS2 was also with the worst prognosis (Fig S10j) as well as fewer BRAF mutations (Fig S10k).

Figure 7. Validation of the subtypes by both experimental and computational methods

a. Scatter plot of the differential expressions of BRAF-relevant genes in CS2 and CS3 subtypes. The x and y axes respectively represent the significant Log2FC calculated for comparing CS2 or CS3 to the other clusters, and only genes showed significant differential expression between PTC tumor

tissue samples with and without BRAF mutations were considered. Significance: Wilcox-test, $P < 0.05$.

b. Immunoblotting analysis with indicated antibodies in IHH4 cells transduced with two independent BRAF shRNAs.

c. Immunoblotting analysis of indicated proteins in BRAF knockdown IHH4 and LY6K transduced cells.

d. Proliferation assay of indicated cell lines ($n=5$).

e-f. Xenograft tumor progression (d) and tumor weight (e). Mice were injected subcutaneously with IHH4 cells transduced with BRAF shRNA alone or in combination with LY6K ($n=5$).

g. Bar plot of the top-10 differentially expressed metabolites between shBRAF2+LY6K and shBRAF2 cells ($n=5$). T-test, $P < 0.01$, rank by P.

h. KEGG based GSEA results for comparing the transcriptomics between different cell lines ($n=5$). Only pathways with the top-10 NES ($P < 0.05$) for comparing shBRAF2+LY6K VS shBRAF2 are shown. meta.: metabolism; inter.: interconversions.

i. Receiver operator characteristic (ROC) curve for predictions of CS1 to CS4 subtypes in the testing dataset.

j. KM-plot of the over all survival curves of the four predicted subtypes in the TCGA-PTC dataset. P: Log-rank test.

Results d-f are the mean of biological replicates from a representative experiment, and error bars indicate s.e.m. Statistical significance was determined by T-test (NS= not significant). The experiments were repeated at least three times.

Figure S10. Validation of the subtypes by both experimental and computational methods

- a. Bar plot of the mean expression of LY6K in PTC tumor tissues with mutant (Mut: N=46) and wild type (WT: n=45) BRAF. P: T-test.
 - b. Bar plot of the mean expression of LY6K in the CS2 (n=15) and CS3 (n=34) subtypes. P: T-test.
 - c-d. Immunoblotting analysis with indicated antibodies in IHH4, TPC1 and BCPAP cells. The experiments were repeated at least three times.
 - e. Diagram of excised tumors from indicated treatment IHH4 cells injected mice (Fig 7e, f) (n=5).
 - f. Bar plot of the differentially expressed metabolites between different conditions. The left-most column was based on the PTC tumor tissues and the other two columns were based on the cell lines. P: Wilcox-test.
 - g-h. Scatter plot showing the feature importance of the top-30 genes (g) and top-30 metabolites (h) for the subtype predictor. The MeanDecreaseGini was estimated based on the random forest algorithm.
 - i. KM-plot of the recurrence free survival curves of the PTC patients predicted with high and low CS2 probabilities (threshold was set as 0.25, since there were four subtypes). P: Log-rank test.
 - j. KM-plot of the overall survival curves of the predicted CS2 and other subtypes in the TCGA-PTC dataset. P: Log-rank test.
 - k. Bar plot of the proportion of BRAF statuses across the four predicted clusters in the TCGA-PTC dataset.
- Results in a-b are the mean values, and error bars indicate standard error.

Major concerns

1) How is the tissue is collected? Do the authors control warm and cold ischemia ? No details on the SOPs employed for the collection, processing and biobanking of tissue is provided. I don't find details on the QC protocol for tissue – tumour content and biomarkers employed for the authentication of tissue samples

Re: We added more descriptions about sample collection, in Line 516-526, Page 34, as below:

The consecutive samples used for this study were selected from patients diagnosed with PTC from Oct 2014 to Jul 2021 at Fudan University Shanghai Cancer Center (FUSCC) in China. The sample collection, store and quality control were in accordance with the standard operation procedures of the Institutional Tissue Bank (ITB) of FUSCC. As described in the previous study [23], after the samples were detached from the human body, they were stored in liquid nitrogen within 30 minutes, and they were made into frozen sections and paraffin-embedded sections at the same time, which were then stained by hematoxylin and eosin. All hematoxylin and eosin slides of the samples were subjected to evaluation for histopathological morphology and tumor components by expert pathologists. The samples enrolled in this study should meet the following criteria: (1) the percentage of tumor cell nuclear (tumor cell nuclear/total cell nuclear) $\geq 80\%$, (2) the percentage of total cells $\geq 80\%$ (cell area/ total tissue section area) and (3) the percentage of necrosis $\leq 20\%$ (necrotic tissue area/total tissue section area).

2) It would be nice to have a table mentioning the range of age, gender, tumor stage, additional therapies or comorbidities for better overview.

Re: We have added the information in Table 1 and Table S1.

3) For figure 1, Here you plot tumor stages for specific target genes. Did you check for pathways or genes that are expressed differently in early and later tumor stages?

Re: Taken Fig 1c for example, we group the patients into different tumor stages, and then for each stage (like T1 stage), we counted how many samples were with mutated MUC16 and how many samples were with wild-type MUC16, and calculated the percentage. However, our study mainly focused on the biological features of different types of recurrence risks, and we did not check for pathways or genes that are expressed differently in early and late tumor stages.

4) Figure 2: There is a low overlap between the transcriptomic with proteomic pathway: did you check for protein modifying enzymes or regulatory pathways responsible?

Re: Similar low overlap between transcriptomics and proteomics was also observed by multiple previous studies:

- ✧ Petralia, F. et al. Integrated proteogenomic characterization across major histological types of pediatric brain cancer. *Cell* 183, 1962–1985.e31 (2020).
- ✧ Shi, X. et al. Integrated proteogenomic characterization of medullary thyroid carcinoma. *Cell Discovery* 8:120. (2022)
- ✧ Huang, C. et al. Proteogenomic insights into the biology and treatment of HPV-negative head and neck squamous cell carcinoma. *Cancer cell* 39: 361-379.e16 (2021).

Based on these studies, we made some additional analysis about the low RNA-protein correlations. This part was added in Page 17, Line 244-256, and Fig S6, as below:

The proteomics are expected to be highly correlated with the transcriptomics, according to the central dogma that the information passes from DNA to RNA to protein. However, relatively low correlations were observed between transcriptomics and proteomics or phospho-proteomics (Fig 4a). From the basis, the mRNA-protein correlations were low, with the sample-wise and gene-wise median mRNA-protein Spearman correlation coefficients respectively 0.29 and 0.086 (Fig S6a-S6b). From the perspective of pathways, the genes/proteins of metabolism pathways showed a relatively higher correlations (but still around 0.25~0.5) than other pathways, while the genes/proteins involved in pathways with large protein complexes like ribosome, spliceosome, mRNA surveillance and autophagy, displayed low or even opposite correlations (around -0.25~0) (Fig S6c, Table S2). Similar results were also reported by previous studies [21, 27, 28]. Besides, dys-regulation in post-transcriptional modifications, like ubiquitination enzymes (HUWE1, CUL4A, TRIM25, etc.) and deubiquitination enzymes (USP7, USP10, USP24, etc.) in these PTC tumor samples (Fig S6d) may also lead to the low consistency.

Fig S6. Low correlations between mRNAs and proteins.

a-b. Histogram of sample-wise (a) and gene-wise (b) mRNA-protein correlations. For the correlation analysis, only mRNAs and proteins that can be recognized by the same gene symbol names were considered, and only samples measured by both transcriptomics and proteomics were used. For the sample-wise correlations, each individual sample was described by two vectors that respectively represented the expressions of all available mRNAs and the corresponding proteins in this sample, and Spearman correlation coefficients between the two vectors were calculated. For the gene-wise correlations, each gene was described by two vectors that represented the expressions of this gene matched mRNA and protein across all the samples, and Spearman correlation coefficients between the two vectors were calculated.

c. KEGG pathways enriched for higher or low gene-wise mRNA-protein correlations (GSEA, $P < 0.05$).

d. Boxplot of the protein expressions of E3 ubiquitin ligases or deubiquitinating enzymes in the normal and tumor samples. (Wicox-test).

5) Fig 2: Could you identify mutations in any enzymes related to the metabolic pathways in tumor samples?

Re: We have added analysis about the relationships between the three most frequent gene mutations and pathway alterations, and found that each of the mutation can lead to alterations in metabolism processed to some degree. The detailed information was added in Page 13, Line 182-187, as below:

Based on these multi-omics data, we also observed the potential impacts of the three most frequent mutations BRAF, MUC16 and TERT promoter on the multi-omics profiles (Fig S4a-b). Enrichment analyses showed these mutations were also related with alterations in multiple metabolism pathways, especially for BRAF (Fig S4c). For example, the differentially expressed mRNAs between PTC samples with and without BRAF mutations were significantly enriched by glycerolipid metabolism and biosynthesis of amino acid pathways (Fig S4c).

Fig S4. Multi-omics alterations related with gene mutations in *BRAF*, *MUC16* and *TERT* promoter.

a. Molecules with significant differential expressions in patients with and without mutations in *BRAF*, *MUC16* or *TERT* promoter (Wilcox-test, $P < 0.05$ and top-10 rank by Log_2-FC in each type of omics for each mutation).

b. Metabolites with significant differential abundances in patients with and without mutations in *BRAF*, *MUC16* or *TERT* promoter (Wilcox-test, $P < 0.05$ and top-10 rank by Log_2-FC for each mutation).

c. KEGG pathways enriched by the molecules with significant differential expressions (Wilcox-test, $P < 0.05$) in patients with and without mutations in *BRAF*, *MUC16* or *TERT* promoter (pathway enrichment examined by hypergeometric distribution, $P < 0.05$ and top-10 rank by P).

6) The authors claim that some of these metabolite levels has been confirmed in the serum levels of the patients: could you please provide the data for their measurement in the serum?

Re: Here we mentioned about the serum level of Tg and PLG in PTC samples **according to several previous literatures(reference [3] and [31]), and did comparisons to our results**, as below:

Plasminogen (PLG) was reported to show significantly lower expressions in serum samples of PTC patients than the nodular goiter patients [31]. Here, the mRNA expression levels of *PLG* were higher in the low RR dominated subtype CS1 than the other subtypes (Fig. 6a).

It has been reported that elevated post lobectomy serum Tg can be used to predict high recurrence risk [3]. However, the other information about the molecular characteristics of different RRs in PTC was limited. Here, we found the high RR was associated with elevated levels in triglycerides, genes *MMP13* and *CST1*, proteins Tg, PTPRG and VWA1 and phosphorylated EPPK1, ALDH1A1 and LAMC1, etc.

We did not perform any measurements in the serum.

Minor

1. Legend of Fig S1: basically the whole legend is wrong:the legend for S1a does not match the graph, I think the legend is for S1b and they just *forgot S1a?

Re: We have revised Fig S1 and the legend.

2 . in S1b legend, there is a typo

Re: We have revised this.

3. n S1c legend "comparision"

Re: We have revised this.

4. Fig 1a: They should organize the legend in the graph in a better way. It is kind of all over the place, just put it in the same order as the graph above.

Re: We have re-organized the legend in the graph

5. Fig 1b: legend for the * is missing. $p < 0.05$?

Re: Yes. We have noted this in the legend.

6. Fig 1 c&d: Sometimes they annotate recurrence risk as "RR" and sometimes as "RecurRisk", they should decide on one

Re. We have revised the RecurRisk into RR.

7. Row 101-106 in the text, they talk about mutations enriched in the high RR patients and cite Fig 2b and 2d but those figures show differentially expressed molecules (DEMs). Am I missing something here?

RE: Sorry. We made a mistake. We have revised Fig 2b and 2d into Fig 1b and 1d.

8. Fig S4 only has legends for 2 of the 5 letters in the graph

RE: We have revised the legends as below:

Fig S5. Recurrence risk relevant molecular and pathway characteristics.

a-d. The differential expressions between tumor and normal samples for the recurrence risk relevant metabolites (a), mRNAs (b), proteins (c) and phospho proteins (d).

e. The GSEA pathway enrichment results for PTC with different recurrence risks in addition to metabolism pathways. The results for metabolism pathways were included in Fig 3e. *: $P < 0.05$, by GSEA.

9 Row 225-228 in the text makes no sense? something is missing

Re: We have revised this part, Line 237-239, as below:

The correlations between different types of omics data were evaluated based on a supervised multi-omics integrative analysis method called DIABLO (Data Integration Analysis for Biomarker discovery using Latent cOmponents) [23]

10. In Fig S5b remove one "the".

Re: We have revised this.

11. Fig S6 legend a: differet was written instead of different

Re: We have revised this.

12. Fig S7 legend a: more description of the analysis is needed, was this done from the Drug Gene Interaction Database?

Re: We have added the description as below:

Fig S9. Potential targets of the four subtypes.

a. Druggable targets of the four subtypes. Druggable targets among the top-10 ranked subtype associated genes were selected based on the DGIdb database.

13. The supplemental Figure 7 (S7) was not cited at all in the paper.

Re: We have added citations of Fig. S9 (original Fig S7) as below:

Candidate druggable targets for each subtype were identified (Fig S9a-b).

Reviewer #2 (Remarks to the Author): Expert in thyroid cancer metabolism

The work of Ning Qu and colleagues is based on an extensive molecular profiling (multi-omics integrated analysis) of papillary thyroid carcinoma samples (PTCs) displaying – following ATA guidelines – different recurrence risks (RR). The work identified distinctive and relevant molecular features of PTCs, allowing to redefine the 3 RRs categories (low, middle, high risk) into four distinct molecular subtypes based on genomic, transcriptional, proteomic and metabolic data. The work is of high relevance to the field and it also represents a potential guideline for a better characterization of RR in other tumors. The paper is well-written, and the methods reported herein are well described, allowing to reproduce the study.

Overall, the results of the work support the conclusions even though, as detailed in the comments below, I have some concerns and suggestions to hopefully improve the overall quality of the manuscript.

Re: Thanks a lot for the affirmation of our work and the valuable suggestions. We have tried our best to revised the manuscript according to the concerns and suggestions.

Major:

1) In the genomic analysis of somatic mutations in PTC samples, the authors do not identify mutations in RAS genes, considering that in the TCGA collection, as well as in many other large or small cohorts (GENIE v3 among them), the cumulative mutation frequency for RAS genes (H-, N- and K-RAS) is even 10-15%. Can this be explained only by geographical reasons? How the authors explain this discrepancy? Did the authors check – by standard Sanger sequencing – some of the PTC samples that were negative for most of the known mutations (at least BRAF and RET/PTC fusions)?

Re: Thanks a lot for this suggestion. We did more investigations about the RAS mutations.

According to the TCGA-PTC analysis, RAS mutation frequency was about 13%.

However, according to the mutational profiling in Chinese patients, the RAS mutation frequency was about 4.1% -6%.

- ✧ **4.1%**: Du Y, et al. Mutational profiling of Chinese patients with thyroid cancer. *Front Endocrinol.* 2023;14:1156999.
- ✧ **6%**: Li, M., et al., Genomic characterization of high-recurrence risk papillary thyroid carcinoma in a southern Chinese population. *Diagnostic Pathology*, 2020. 15(1).

Consequently, the Chinese PTC patients had fewer RAS mutations indeed.

In our study, the RAS mutation frequency was 0. The none RAS mutation may be caused by not only the geographical reason but also the limitation of sample collection. We added discussions about this issue in Line 106-107, Page 7:

Meanwhile, this Chinese PTC cohort did not contain mutations in RAS which was mutated in 13% of samples for the TCGA-PTC cohort [10], but the mutation frequency was about 4.1% - 6.0% in the other Chinese cohorts [16, 17].

and Line 456-458, Page 31:

RAS mutations in Chinese PTC cohorts [16, 17] were much less than that reported by TCGA [10]. Here, no RAS mutation was identified, probably also due to the geographical as well as sample limitation.

Beside, in the beginning of this study we parallelly did sanger sequencing for several PTC samples to confirm the high throughput sequencing result and the BARF mutation 100% matched in two different methods, but we did not involve sanger sequencing results here. We also analyzed the TERT promoter mutation frequent by sanger sequencing.

2) The frequency of TERT promoter's mutation is highly variable in PTC. It ranges from about 30%, as observed by the GENIE consortium, to very low frequencies such as in TCGA and in a work from Nikiforov's group (Panebianco et al., 2019) that found this mutation in 7% of PTC cases. As this mutation does not fall into the WES panel, the authors used ARMS-qPCR to detect it. Did the authors validate (here or in the previous work) the accuracy of this detection system? Is there the possibility that the measured percentage is biased by the detection method, or it reflects the mutations frequency in Chinese PTCs? Are there other works on similar (by geographical distribution) populations regarding PTCs? If so, please include in the manuscript and discuss on it.

Re: The ARMS-qPCR method was validated in a previous study (Yu PC, Tan LC, Zhu XL, et al. Arms-qPCR Improves Detection Sensitivity of Earlier Diagnosis of Papillary Thyroid Cancers With Worse Prognosis Determined by Coexisting BRAF V600E and Tert Promoter Mutations. *Endocr Pract.* 2021;27(7):698-705. doi: 10.1016/j.eprac.2021.01.015.)

The prevalence of TERT C228T promoter mutations in Chinese PTC varied between 4.1%–9.6% according to previous studies as below:

- ✧ **9.6%**: Liu X, Qu S, Liu R, et al. TERT promoter mutations and their association with BRAF V600E mutation and aggressive clinicopathological characteristics of thyroid cancer. *J Clin Endocrinol Metab.* 2014 Jun;99(6):E1130-6. doi: 10.1210/jc.2013-4048).
- ✧ **5.2%**: Du Y, Zhang S, Zhang G, et al. Mutational profiling of Chinese patients with thyroid cancer. *Front Endocrinol.* 2023;14:1156999. doi: 10.3389/fendo.2023.1156999.
- ✧ **4.1%** (when only tumors ≥ 1.5 cm were analyzed in the present cohort of PTC, the prevalence of TERT promoter mutations was 9.8%): Jin L, Chen E, Dong S, et al. BRAF and TERT promoter mutations in the aggressiveness of papillary thyroid carcinoma: a study of 653 patients. *Oncotarget.* 2016 Apr 5;7(14):18346-55. doi: 10.18632/oncotarget.7811.

Besides, the higher TERT C228T mutation frequency (14%) in our cohort may also be related to the larger number of high-recurrence risk samples. Since the TERT C228T mutation was significantly associated with high recurrence risk in PTC (Agrawal N, Akbani R, Aksoy BA, et al. Integrated genomic characterization of papillary thyroid carcinoma. *Cell*. 2014 Oct 23;159(3):676-90. doi: 10.1016/j.cell.2014.09.050.).

We added additional discussions about this in Line 458-462, Page 31-32, as below:

The prevalence of TERT C228T promoter mutations in Chinese PTC varied between 4.1%–9.6% according to previous studies [17, 36, 37]. The high TERT C228T mutation (14%) frequency here may be related with the large proportion of high RR patients (n=48, 47.06%) in the cohort, since the TERT C228T mutation was found significantly associated with high RR in PTC.

3) Considering the relevant number of samples analyzed by transcriptome analysis, did the authors identify completely new gene rearrangements (searching for fusion/chimeric transcripts)? If so, please describe them and also verify the expression of related parent genes in the same data sets, possibly commenting on the findings in terms of oncogenic/tumor-suppressive potential of the new chimeric transcripts.

Re: We have already identified the gene fusions based on the transcriptome data in Line 114-120, Page 8, as below:

Here, RET fusions (CCDC6-RET 8%, NCOA4-RET 5%) were the most frequent fusions, and multiple NTRK fusions (NTRK3-ETV6, TPR-NTRK1, ETV6-NTRK3) were also identified (Fig 1a, Fig S1c). In addition, several other gene fusions (FBXO25-SEPTIN14, TLK2-FAM157A, ZNF33B-NCOA4) showing rare frequencies in previous PTC studies were also identified (Fig 1a). Interestingly, several gene fusions (NCOA4-RET, TLK2-FAM157A, ZNF33B-NCOA4, TPR-NTRK1) also showed specific enrichment in the high RR (Fig 1b).

However, the oncogenic/tumor-suppressive potential of the gene fusions should be determined by more systemic analysis, and not discussed in this study. We will characterize these new transcripts oncogenic/tumor-suppressive functions in next study.

4) In the integrated analysis the authors focus on common pathways affected. How do the authors explain the low overlap - in terms of enriched pathways - among transcriptome, proteome and phosphoproteome? Is it due to the low correlation between single omic data sets (transcriptome vs proteome, proteome vs phosphoproteome etc)? If so, could it be relevant to consider also the “application-specific” ones? Could this discrepancy reflect an unknown biological/pathological factor rather than only depend on “technical” variability?

Re: Similar low overlap between transcriptomics and proteomics was also observed by multiple previous studies:

- ✧ Petralia, F. et al. Integrated proteogenomic characterization across major histological types of pediatric brain cancer. *Cell* 183, 1962–1985.e31 (2020).
- ✧ Shi, X. et al. Integrated proteogenomic characterization of medullary thyroid carcinoma. *Cell Discovery* 8:120. (2022)

- ✧ Huang, C. et al. Proteogenomic insights into the biology and treatment of HPV-negative head and neck squamous cell carcinoma. *Cancer cell* 39: 361-379.e16 (2021).

Based on these studies, we made some additional analysis about the low RNA-protein correlations. This part was added in Page 17, Line 244-256, and Fig S6, as below:

The proteomics are expected to be highly correlated with the transcriptomics, according to the central dogma that the information passes from DNA to RNA to protein. However, relatively low correlations were observed between transcriptomics and proteomics or phospho-proteomics (Fig 4a). From the basis, the mRNA-protein correlations were low, with the sample-wise and gene-wise median mRNA-protein Spearman correlation coefficients respectively 0.29 and 0.086 (Fig S6a-S6b). From the perspective of pathways, the genes/proteins of metabolism pathways showed a relatively higher correlations (but still around 0.25~0.5) than other pathways, while the genes/proteins involved in pathways with large protein complexes like ribosome, spliceosome, mRNA surveillance and autophagy, displayed low or even opposite correlations (around -0.25~0) (Fig S6c, Table S2). Similar results were also reported by previous studies [21, 27, 28]. Besides, dys-regulation in post-transcriptional modifications, like ubiquitination enzymes (HUWE1, CUL4A, TRIM25, etc.) and deubiquitination enzymes (USP7, USP10, USP24, etc.) in these PTC tumor samples (Fig S6d) may also lead to the low consistency.

Fig S6. Low correlations between mRNAs and proteins.

a-b. Histogram of sample-wise (a) and gene-wise (b) mRNA-protein correlations. For the correlation analysis, only mRNAs and proteins that can be recognized by the same gene symbol names were considered, and only samples measured by both transcriptomics and proteomics were used. For the sample-wise correlations, each individual sample was described by two vectors that respectively represented the expressions of all available mRNAs and the corresponding proteins in this sample, and Spearman correlation coefficients between the two vectors were calculated. For the gene-wise correlations, each gene was described by two vectors that represented the expressions of this gene matched mRNA and protein across all the samples, and Spearman correlation coefficients between the two vectors were calculated.

c. KEGG pathways enriched for higher or low gene-wise mRNA-protein correlations (GSEA, $P < 0.05$).

d. Boxplot of the protein expressions of E3 ubiquitin ligases or deubiquitinating enzymes in the normal and tumor samples. (Wicox-test).

5) When describing the metabolic alterations that the authors identified in PTC (lines 168-172), other - and more relevant than ref. 9 (line 169) - citations could be added to further highlight that multiple independent studies have already identified gene/protein signatures and metabolic changes in glucose uptake, glycolysis and lipid metabolism in PTCs (Chai et al. Surgery 2017; Ma et al. J Clin Endocrinol Metab 2019; Ban EJ et al. Endocrinol Metab 2021; Xu F et al. Front Oncol 2021; Wen et al. Acta Biochim Biophys Sin 2021; Aprile et al., Br J Can 2023). Noteworthy, some of these recent works reported a correlation with poor prognosis, tumor relapse and/or low therapeutic response. These works may be also briefly commented in the Discussion (around line 399) to foster an open field represented by the targeting of metabolic pathways, also in the PTC context.

Re: Thanks a lot for the valuable suggestion, and we have added these references as below:

Line 62-63, Page 4:

Transcriptomics based analysis also identified various metabolic enzymes that play key roles in PTC [12-15].

Line 470-472, Page 32:

Previous studies have also identified metabolism factors involved in glucose uptake, glycolysis and lipid metabolism showing significant associations with PTC prognosis [12, 13, 42] or therapeutic responses [43].

6) Considering that transcriptome and proteome data indicate immune mediators and immunity-related pathways/processes as differentially expressed, and the omic analyses have been done on bulk tissue biopsies, could the different cell composition - reported on lines 335-338 - depend on the heterogeneity of tumor biopsies?

Re: Yes. Here the compositions of different cell types in each bulk sample were estimated based on the previously reported single-cell RNA-seq data of PTC tumor samples, and the cell composition differences among the PTC samples were related with the heterogeneity of tumor samples.

7) The authors identified by multi-omic integrated cluster analysis the presence of 2 BRAF-like subgroups having completely opposite RR (ie. low and high). Considering the heterogeneous landscape of somatic alterations in the BRAF-mutated PTCs, did the authors check if CS4 cluster is enriched in other (additional) somatic mutations (TERT or other MAPK or PI3K genes) and/or other gene fusions that may account for the completely different RR?

Re: We checked the other somatic mutations and gene fusions, and CS4 was also enriched with TERT mutation ($p = 0.02937$, Chi-squared Test). We added this result in Line 310-311, Page 21, as below:

The subtype CS4 was significantly enriched by BRAF and TERT promoter mutations (P = 0.007227 for BRAF, P= 0.02937 for TERT, Chi-squared Test, Fig 5b).

8) Considering the low correlation, as measured by the authors, between transcriptome and proteome/phosphoproteome data in these PTC samples, and considering also that proteins, rather than transcripts, are the bona fide targets of drugs, it would be expected to build a network using proteome/phosphoproteome data rather than mRNA expression. Could the authors comment on this and adjust their analysis making it more focused on deregulated proteins/enzymes?

Re: This was a good suggestion. However, since the subtypes were identified based on the mRNA and metabolism data (Fig 5), we directly utilized the mRNA data to identify potential subtype-relevant targets (Fig S9).

The mRNA and metabolism data were utilized for subtype identification considering two main aspects. Firstly, the proteomics/phosphoproteome showed high correlations with metabolism data (Fig.4), thus may lead to data redundancy for clustering algorithm, so the proteomics/phosphoproteome were not utilized for the subtype identification. Secondly, the number of genes in the mRNA data was much larger than the available number of proteins measured by the proteome, thus covering more druggable targets.

On the other hand, although low correlation values between transcriptomics and proteomics in general, the mRNA data can still provide clues on potential targets as did in the other studies:

- ✧ Ding RB, Chen P, Rajendran BK, et al. Molecular landscape and subtype-specific therapeutic response of nasopharyngeal carcinoma revealed by integrative pharmacogenomics. *Nat Commun.* 2021;12(1):3046;
- ✧ Elango R, Rashid S, Vishnubalaji R, et al. Transcriptome profiling and network enrichment analyses identify subtype-specific therapeutic gene targets for breast cancer and their microRNA regulatory networks. *Cell Death Dis.* 2023;14(7):415;
- ✧ Neff RA, Wang M, Vatansever S, et al. Molecular subtyping of Alzheimer's disease using RNA sequencing data reveals novel mechanisms and targets. *Sci Adv.*;7(2):eabb5398. doi: 10.1126/sciadv.abb5398.)

Besides, the network considering all four types of omics (mRNA, metabolism, proteome/phosphoproteome) was already built in Fig.4c. So, we revised the original Fig 7 as Fig S9, and did not use the mRNA-metabolite correlations to indicate metabolite biomarkers.

9) At the end of Discussion the authors comment on the most relevant result of their work, ie. the conclusion that ATA stratification may not be taken as the only single predictor for PTC. It is supported by the findings of the authors. However, how do the authors propose to employ their subclassification based on multiple omics and bioinformatics in a routine clinical practice that is based on the management of hundreds/thousands of PTC patients in clinics or hospitals? Did the authors propose a minimal subtype-specific "signature" of molecules to be tested (eg. a panel made by a specific combination of x genes, y proteins and/or z metabolites etc.) that can be eventually used in the clinical practice with reasonable low costs and in a reasonable amount of time? It would provide a significant clinical impact and would strengthen the current work.

Re: We did build a subtype predictor based on the top-30 important genes and top-30 important metabolites (see Methods : Subtype prediction). Based on the current data, the transcriptomics and metabolomics data of the 97 PTCs were partitioned into training (60%) and testing datasets (40%). The test results show this predictor has a good performance (see Fig 7i) and the predicted CS2-score can help predict prognosis as well (Fig S10i).

This machine-learning based subtype predictor only used the most important 30 genes and 30 metabolites (as shown in Fig S10g-h), providing a potential manner that can be used in clinical practice in the future.

Fig S10. g-h. Scatter plot showing the mean decrease Gini of the genes (a) and metabolites (b). The MeanDecreaseGini was estimated based on the random forest algorithm.

Minor:

1) The discrepancy of the results on the genomic analysis (compared to TCGA or other large consortia) regarding MUC16 may be due to the filtering process adopted by genomic centers, as some of these genes (including also RNF213) are often considered frequently-mutated “not-specific” cancer genes.

Re: Yes. We added some discussion about this issue in Line 466-468, Page 32:

Notably, the discrepancy of the identified mutations compared to TCGA and other cohorts may also probably due to the different mutation filtering strategies.

2) In the paragraph “Multi-omics based comparison of tumor and normal tissues of PTC patients”, I suggest removing the sentences with "etc." (lines 125-7) describing the genes/proteins differentially expressed. These sentences, in my opinion, do not add to the manuscript.

Re: Yes. We have removed the sentences.

3) Lines 129-130 Fibronectin 1 is repeated two times.

Re: We have revised this.

4) Lines 190-192. The authors found PTC tumor samples having lower Tg compared to normal samples, but “high RR” PTC samples behave the opposite compared to other tumor subtypes. What could be the explanation for this?

Re: The comparisons were based on different sample types. The tumor samples (all different RR together) were with low Tg levels than the normal samples. Meanwhile, among all the PTC tumor samples, the high RR PTC tumor samples were with higher Tg levels than the other tumor tissues.

The data distribution for the high-RR, median-RR, low-RR and normal tissues were as below:

Reviewer #3 (Remarks to the Author): Expert in MS-based proteomics, phosphoproteomics, and metabolomics

Although thyroid cancer is considered a rare cancer that affects the thyroid gland, the estimated global incidence rate is rising quickly. Age, gender, and radiation exposure are a few variables that may increase the risk of thyroid cancer. Recurrence risks (RRs) is categorized into three levels risk stratification: low, intermediate, and high. However, the molecular underpinnings of the various RRs remain poorly known. The identification of biomolecules that may contribute to a better PTC risk stratification has utmost importance for the development of effective diagnostic and treatment strategies. Within this context, the topic of this study is timely and pertinent. Specifically, Qu et al performed a integrated and comprehensive multiomics analysis to identify molecular determinants associated with the four different type of PCT RRs cancer. Overall the text is well structured and the data gathered is sound and support most of the claims made throughout the text. However this reviewer has major points that need to be address by the authors.

1) The authors should state clearly the study's calculating power because 102 samples seemed insufficient for this type of research. This brings me to my second point 2). Although the authors conducted a thorough integrated multiomics analysis, the work is primarily descriptive and to some extent lacks validation or verification. The authors should considered targeted approaches (metabolomics/proteomics) to confirm potential biomarkers/druggable targets claimed to be associated with the various types of PCT RRs cancer . Such a strategy might strengthen the study's conclusions and minimize the impacts of the small sample size.

Re: Although the current sample size was 102, the multi-omics data based on the current sample size have generated a great deal of statistically significant results as displayed in the manuscript. Besides, considering to validate the finally redefined subtypes in our study, we developed a subtype predictor based on our data, and validated it in the TCGA-PTC cohort (n=505), as we have added in Line **:

Meanwhile, the sample sizes for multiple previous multi-omics studies were also around 100, as below:

- ✧ Herbst SA, Vesterlund M, Helmboldt AJ, et al. Proteogenomics refines the molecular classification of chronic lymphocytic leukemia. *Nat Commun.* 2022 Oct 20;13(1):6226. doi: 10.1038/s41467-022-33385-8. PMID: 36266272 (n=68)
- ✧ Wang LB, Karpova A, Gritsenko MA, et al. Proteogenomic and metabolomic characterization of human glioblastoma. *Cancer Cell.* 2021 Apr 12;39(4):509-528.e20. doi: 10.1016/j.ccell.2021.01.006. Epub 2021 Feb 11. PMID: 33577785 (n=99)
- ✧ Shi X, Sun Y, Shen C, et al. Integrated proteogenomic characterization of medullary thyroid carcinoma. *Cell Discov.* 2022 Nov 8;8(1):120. doi: 10.1038/s41421-022-00479-y. PMID: 36344509; PMCID: PMC9640541. (n= 102)
- ✧ Krug K, Jaehnig EJ, Satpathy S, et al. Proteogenomic Landscape of Breast Cancer Tumorigenesis and Targeted Therapy. *Cell.* 2020 Nov 25;183(5):1436-1456.e31. doi: 10.1016/j.cell.2020.10.036. Epub 2020 Nov 18. PMID: 33212010; PMCID: PMC8077737. (n=122)

✧ Cao L, Huang C, Cui Zhou D, et al. Proteogenomic characterization of pancreatic ductal adenocarcinoma. *Cell*. 2021 Sep 16;184(19):5031-5052.e26. doi: 10.1016/j.cell.2021.08.023. PMID: 34534465.(n=140)

We also added some experimental and computational assays to validate some key findings, as added in Page 27-31, as below:

Validation of BRAF-status relevant subtype characters in PTC cells

The BRAF-status was highly correlated to different subtypes of PTC (Fig 5a-b and Fig 6g). Meanwhile, for the two subtypes CS2 and CS3 that with fewer BRAF mutations but high recurrence rates, our investigation found that they showed opposed alterations in some BRAF-mutation relevant genes, such as LY6K (Fig 7a, Fig S10a-b). To further validate these correlations, we initially examined the BRAF and its mutant form, BRAFV600E, in different PTC cells (Fig S10a). Notably, IHH4 cells exhibited abundant expression of both BRAF and BRAFV600E in comparison to other PTC cells (Fig S10c). To interfere with both BRAF and BRAFV600E expression, we introduced two independent BRAF short-hairpin RNAs (shRNA) into IHH4 cells, resulting in a simultaneous reduction in the expression of BRAF and BRAFV600E (Fig 7b). Then, we investigated the expression of LY6K (Fig S10d). We observed that LY6K expression decreased alongside BRAF/BRAFV600E downregulation, indicating a potential reliance on BRAF (Fig 7c). Further exploration revealed that the restoration of LY6K significantly reversed cell proliferation and tumorigenesis in BRAF knockdown cells (Fig 7d-f and S10e). These results suggest that the BRAF-status may interact with other factors, such as metabolic signaling, within PTC, thus cooperatively contributing to the four distinctive subtypes to a certain degree.

Next, to validate the correlation between BRAF-status and metabolic signaling, we conducted metabolomics and transcriptomics analyses using the aforementioned cell lines. Consistent with the observation that multiple metabolites showed increased levels in BRAF-mutant PTC tumor samples (Fig S4b), the controlled IHH4 cells also showed improved levels in a series of metabolites, especially PE and PC species, comparing to the BRAF knockdown cells (Fig S10f). The restoration of LY6K lead to significantly decreased levels in multiple metabolites (Fig 7g) and up-regulations of genes in some metabolism pathways like drug metabolism and pentose and glucuronate interconversions (Fig 7h), in agreement with the reduction of most metabolites and up-regulation of genes involved in various metabolism pathways for the LY6K-high subtype CS2 (Fig 6b, Fig 6h). Meanwhile, the restoration of LY6K also reversed the pathway impacts generated by BRAF knockdown (Fig 7h).

Validation of the subtypes by machine learning models

To validate the four PTC subtypes *in silico*, we splited the PTC data into training and testing datasets and constructed a subtype predictor based on the expression levels of the top-30 ranked genes and metabolites in the training dataset (Fig S10g-h). The predictor can classify the four subtypes accurately in the testing dataset, with the areas under the receiver operator characteristic curves (AUCs) 1, 0.97, 0.97, 0.99 for CS1 to CS4 on the testing dataset (Fig 7i). Meanwhile, the predicted probabilities of being the subtype CS2 were associated with recurrence-free survival where high CS2 probabilities were associated with unfavourable prognosis (Fig S10i). Moreover, to validate the subtype by external cohorts where only transcriptomics were available, we also trained a subtype predictor based on the top-300 mRNA expressions (mean accuracy across 10-fold cross-over validations: 0.8234), and utilized it to find the four subtypes in the TCGA-PTC cohort (see Methods).

Subtypes with similar prognosis patterns can still be recognized (Fig 7j), where the subtype CS2 was also with the worst prognosis (Fig S10j) as well as fewer BRAF mutations (Fig S10k).

Figure 7. Validation of the subtypes by both experimental and computational methods

a. Scatter plot of the differential expressions of BRAF-relevant genes in CS2 and CS3 subtypes. The

x and y axes respectively represent the significant Log₂FC calculated for comparing CS2 or CS3 to the other clusters, and only genes showed significant differential expression between PTC tumor tissue samples with and without BRAF mutations were considered. Significance: Wilcox-test, P<0.05.

b. Immunoblotting analysis with indicated antibodies in IHH4 cells transduced with two independent BRAF shRNAs.

c. Immunoblotting analysis of indicated proteins in BRAF knockdown IHH4 and LY6K transduced cells.

d. Proliferation assay of indicated cell lines (n=5).

e-f. Xenograft tumor progression (d) and tumor weight (e). Mice were injected subcutaneously with IHH4 cells transduced with BRAF shRNA alone or in combination with LY6K (n=5).

g. Bar plot of the top-10 differentially expressed metabolites between shBRAF2+LY6K and shBRAF2 cells (n=5). T-test, P<0.01, rank by P.

h. KEGG based GSEA results for comparing the transcriptomics between different cell lines (n=5). Only pathways with the top-10 NES (P<0.05) for comparing shBRAF2+LY6K VS shBRAF2 are shown. meta.: metabolism; inter.: interconversions.

i. Receiver operator characteristic (ROC) curve for predictions of CS1 to CS4 subtypes in the testing dataset.

j. KM-plot of the over all survival curves of the four predicted subtypes in the TCGA-PTC dataset. P: Log-rank test.

Results d-f are the mean of biological replicates from a representative experiment, and error bars indicate s.e.m. Statistical significance was determined by T-test (NS= not significant). The experiments were repeated at least three times.

Figure S10. Validation of the subtypes by both experimental and computational methods

- a. Bar plot of the mean expression of LY6K in PTC tumor tissues with mutant (Mut: N=46) and wild type (WT: n=45) BRAF. P: T-test.
 - b. Bar plot of the mean expression of LY6K in the CS2 (n=15) and CS3 (n=34) subtypes. P: T-test.
 - c-d. Immunoblotting analysis with indicated antibodies in IHH4, TPC1 and BCPAP cells. The experiments were repeated at least three times.
 - e. Diagram of excised tumors from indicated treatment IHH4 cells injected mice (Fig 7e, f) (n=5).
 - f. Bar plot of the differentially expressed metabolites between different conditions. The left-most column was based on the PTC tumor tissues and the other two columns were based on the cell lines. P: Wilcox-test.
 - g-h. Scatter plot showing the feature importance of the top-30 genes (g) and top-30 metabolites (h) for the subtype predictor. The MeanDecreaseGini was estimated based on the random forest algorithm.
 - i. KM-plot of the recurrence free survival curves of the PTC patients predicted with high and low CS2 probabilities (threshold was set as 0.25, since there were four subtypes). P: Log-rank test.
 - j. KM-plot of the overall survival curves of the predicted CS2 and other subtypes in the TCGA-PTC dataset. P: Log-rank test.
 - k. Bar plot of the proportion of BRAF statuses across the four predicted clusters in the TCGA-PTC dataset.
- Results in a-b are the mean values, and error bars indicate standard error.

3) Phosphoproteomics results should clearly indicate the p-site (S/T/Y) identified. Was there an attempt to calculate the occupancy of the p-sites that were identified.

Re: For this study, we only utilized the total phosphorylation levels to do the analysis.

4) Data accessibility: This reviewer was unable to find any indication of data being made available in open repositories like Metabolomic Workbench or PRIDE for proteomics data. This should have been completed and made accessible to reviewers even before submission.

Re: We have submitted all of the raw data to open repositories. The data availability statement was as below:

The raw WES and RNA-seq data of the PTC samples have been deposited in the Genome Sequence Archive [64] in National Genomics Data Center [65], China National Center for Bioinformatics / Beijing Institute of Genomics, Chinese Academy of Sciences (GSA-Human:

<https://ngdc.cncb.ac.cn/gsa-human/browse/HRA005293>;

<https://ngdc.cncb.ac.cn/gsa-human/browse/HRA005382>). The mass spectrometry proteomics and phospho-proteomics data have been deposited to the ProteomeXchange Consortium via the PRIDE [66] with the dataset identifier PXD044900 and PXD045017. The metabolomics data have been deposited to MetaboLights [67] (www.ebi.ac.uk/metabolights/MTBLS3339). Transcriptomics and survival data of TCGA-PTC samples were obtained from Genomic Data Commons via TCGAbiolinks (v 2.5.9).

For the PRIDE, the up-loaded data is private before publication of the corresponding manuscript, and can be accessed with reviewer accounts as below:

Username: reviewer_pxd045017@ebi.ac.uk; Password: 4ijh4m1K

Username: reviewer_pxd044900@ebi.ac.uk; Password: aLJQZPrC

5) A figure demonstrating the experimental design and providing a detailed description of the cohort under study would also be useful. the sampling (tissue), sample extraction, and downstream analysis.

Re: We have added an overview about the study in Fig S1a, as below.

The detailed information about the cohort was described in Table 1 and Table S1.

The sample collection was revised in Line 516-526, Page 34, as below:

The consecutive samples used for this study were selected from patients diagnosed with PTC from Oct 2014 to Jul 2021 at Fudan University Shanghai Cancer Center (FUSCC) in China. The sample collection, store and quality control were in accordance with the standard operation procedures of the Institutional Tissue Bank (ITB) of FUSCC. As described in the previous study [21], after the samples were detached from the human body, they were stored in liquid nitrogen within 30 minutes, and they were made into frozen sections and paraffin-embedded sections at the same time, which were then stained by hematoxylin and eosin. All hematoxylin and eosin slides of the samples were subjected to evaluation for histopathological morphology and tumor components by expert pathologists. The samples enrolled in this study should meet the following criteria: (1) the percentage of tumor cell nuclear (tumor cell nuclear/total cell nuclear) $\geq 80\%$, (2) the percentage of total cells $\geq 80\%$ (cell area/total tissue section area) and (3) the percentage of necrosis $\leq 20\%$ (necrotic tissue area/total tissue section area).

REVIEWERS' COMMENTS

Reviewer #1 (Remarks to the Author):

The authors have revised the study significantly. The functional validation of BRAF-LY6K axis in tumorigenesis did make a significant contribution here.

- 1) Please explain in the manuscript why LY6K has been selected as the target to test here?
- 2) Include loading controls in all blots.
- 3) Probe for p-MEK1/2 at least to functionally check BRAF.
- 4) Ideally the authors should test the approved BRAF inhibitors for the effects

Reviewer #2 (Remarks to the Author):

The manuscript from Ning Qu and colleagues reports an extensive molecular profiling (multi-omics integrated analysis) on papillary thyroid carcinoma (PTC) samples that display – according to the ATA guidelines and classification – different recurrence risk. The main result of the work is the identification of new redefined categories of recurrence risk into four distinct molecular subtypes. The work is based on the integration of genomic, transcriptional, proteomic and metabolic data. In the revised and, in my opinion, largely improved version of the manuscript, the authors included new relevant experimental (on cell lines) and computational (by machine learning) validations that further contribute to strengthen the work. The machine learning-based subtype predictor - using the most important 30 genes and 30 metabolites - provides a potential tool to be used in the clinical practice in the future. The manuscript is of high relevance to the oncology field, especially regarding thyroid carcinomas, but it may also represent a potential guideline for the definition of new recurrence risk categories in other cancers, based on the widely available omic data for most tumor types.

The paper is well-written, and the revised version has been further improved to support the findings. Additionally, the methods have been implemented with the inclusion of other details relevant to increase the reproducibility of the study and the omic data sets have been deposited to public repositories.

Overall, the results of the work support the conclusions and the authors have clarified all my doubts and positively answered to all the concerns.

Reviewer #3 (Remarks to the Author):

All of the reviewer's earlier concerns have been satisfactorily handled by the authors.

REVIEWERS' COMMENTS

Reviewer #1 (Remarks to the Author):

The authors have revised the study significantly. The functional validation of BRAF-LY6K axis in tumorigenesis did make a significant contribution here.

1) Please explain in the manuscript why LY6K has been selected as the target to test here?

Re: The expression of LY6K showed significant difference in PTC tumor tissues with and without BRAF mutations (Figure S10a-b). Meanwhile, for the two subtypes CS2 and CS3 that with fewer BRAF mutations but high recurrence rates, LY6K also showed significant differences between CS2 and CS3 subtypes and this difference was very specific to CS2 (Fig 7a). Consequently, we select LY6K to test here, and we wanted to show the BRAF-status may interact with other factors, such as metabolic signaling, within PTC, thus cooperatively contributing to the four distinctive subtypes to a certain degree.

We also explained this in the manuscript as below:

The BRAF-status was highly correlated to different subtypes of PTC (Fig 5a-b and Fig 6g). Meanwhile, for the two subtypes CS2 and CS3 that with fewer BRAF mutations but high recurrence rates, our investigation found that they showed opposed alterations in some BRAF-mutation relevant genes, such as LY6K (Fig 7a, Supplementary Fig S10a-b).

2) Include loading controls in all blots.

Re: Indeed, the loading controls of Vinculin were added in Fig 7b in the last version (please see below):

3) Probe for p-MEK1/2 at least to functionally check BRAF.

Re: Thank you so much for your suggestion. We checked the expression of p-MEK1/2 and total MEK1/2 and added these results in the revised version. The results were as below:

4) ideally the authors should test the approved BRAF inhibitors for the effects

Re: Thank you for your valuable suggestion. In this study, our main focus is on characterizing papillary thyroid cancer with varying recurrence risks through multi-omics analysis. This aspect holds significant potential for studying clinical therapy. Consequently, we plan to conduct further studies involving approved BRAF inhibitors to explore their efficacy.